

# JT gravity at finite cutoff

Luca V. Iliesiu[1], Jorrit Kruthoff[2], Gustavo J. Turiaci[3] and Herman Verlinde[1]

**1** Joseph Henry Laboratories, Princeton University, Princeton, NJ 08544, USA
**2** Stanford Institute for Theoretical Physics, Stanford University, Stanford, CA 94305, USA
**3** Physics Department, University of California, Santa Barbara, CA 93106, USA

## Abstract

We compute the partition function of $2D$ Jackiw-Teitelboim (JT) gravity at finite cutoff in two ways: (i) via an exact evaluation of the Wheeler-DeWitt wavefunctional in radial quantization and (ii) through a direct computation of the Euclidean path integral. Both methods deal with Dirichlet boundary conditions for the metric and the dilaton. In the first approach, the radial wavefunctionals are found by reducing the constraint equations to two first order functional derivative equations that can be solved exactly, including factor ordering. In the second approach we perform the path integral exactly when summing over surfaces with disk topology, to all orders in perturbation theory in the cutoff. Both results precisely match the recently derived partition function in the Schwarzian theory deformed by an operator analogous to the $T\bar{T}$ deformation in $2D$ CFTs. This equality can be seen as concrete evidence for the proposed holographic interpretation of the $T\bar{T}$ deformation as the movement of the AdS boundary to a finite radial distance in the bulk.

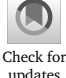

# 1   Introduction

Can the holographic dictionary of AdS/CFT be generalized to gravitational theories defined on a finite patch of spacetime? This question has recently attracted renewed attention due to the discovery of a new class of solvable irrelevant deformations of two-dimensional conformal field theory, known as the $T\bar{T}$ deformation [1–3]. It was conjectured in [4] (see also [5]) that the $T\bar{T}$ deformed CFT can be interpreted as the holographic dual of a finite patch of asymptotically AdS$_3$ spacetime. A key piece of evidence in support of this conjectured duality is that the conformal Ward identity of the CFT gets deformed into a second order functional differential equation that formally matches with the Wheeler-DeWitt equation of AdS$_3$ gravity. This relationship is akin to the familiar duality between Chern-Simons field theory and Wess-Zumino-Witten conformal field theory [6], and points to the possible identification between the wave functionals of gravitational theories in $D + 1$-dimensions and partition functions of a special class of $D$-dimensional QFTs.[1] If such a precise relationship could indeed be established, it would be an important generalization of the standard holographic dictionary that would open up a new avenue for studying gravitational physics in the bulk space-time.

However, while the new duality between AdS$_3$ gravity and the $T\bar{T}$ deformed CFTs passes several non-trivial checks, the precise status of the correspondence is still unclear. In particular, it was found that for large enough energies and fixed deformation parameter, the energy spectrum of the boundary QFT complexifies, indicating a possible breakdown of unitary. This apparent breakdown has a natural interpretation from the point of view of the bulk: it corresponds to a physical cut-off on the spectrum of black hole states, that removes all black holes with Schwarzschild radii that would extend beyond the finite radial cutoff. Nonetheless, the existence of this cut-off in the energy spectrum raises several conceptual questions, that require better understanding of the UV properties of the boundary QFT.

To circumvent the complications of field theory, in this paper we turn to analyzing the

---

[1]For studies of higher-dimensional generalisations of the $T\bar{T}$ deformation and their potential holographic interpretation, see [7, 8].

problem in one dimension lower. In particular we will consider the finite cutoff version of two-dimensional Jackiw-Teitelboim (JT) gravity with a negative cosmological constant and its dual formulation in terms of a deformed version of Schwarzian quantum mechanics proposed in [9, 10]. Traditionally, the JT path integral is computed with Dirichlet boundary conditions in the limit where the proper boundary length and boundary value for the dilaton become very large [11]. In that limit, JT gravity reduces to a soluble 1D quantum theory, the Schwarzian theory [12–14].[2] At finite cut-off, however, the boundary theory is expected to become highly non-linear and the computation of the JT partition function in this regime has thus far been an open problem. In the following, we will discuss two different ways to compute it, relying on either canonical quantization or path integral quantization.

In the canonical approach one foliates the spacetime with a certain, usually time-like, coordinate and parametrizes the metric in an ADM decomposition [28]. As a result of diffeomorphism invariance, the quantum mechanical wave functionals satisfy a set of local Wheeler-DeWitt constraints, that uniquely determine their dependence on local data defined on the chosen foliation. In general these constraints are difficult to solve, except possibly in the so-called mini-superspace approximation. Luckily, for two-dimensional dilaton gravity theories, the constraints reduce to two first order functional differential equations that can be solved exactly [29] [30] in the form of explicit diffeomorphism invariant wavefunctionals of the boundary metric and dilaton profile. The WDW wavefunctionals relevant for our analysis are defined through radial quantisation in Euclidean signature. This provides a way to compute the path integral at finite cutoff.

In the second approach, we compute the Euclidean path integral directly. Again, the analysis at finite proper boundary length becomes more intricate as some of the gravitational modes, that were frozen in the large volume limit, now become dynamical. After integrating out the dilaton, the JT path integral localizes to one over a boundary action given by the extrinsic curvature $K$ of the boundary. By using constraints from the $SL(2, \mathbb{R})$ isometry of AdS$_2$, we manage to express $K$ in an expansion containing solely powers of the Schwarzian derivative and its derivatives. This greatly facilitates our computations and allows us to express the partition function as the expectation value of an operator in the Schwarzian theory. Using integrability properties of the Schwarzian theory, we manage to exactly compute the partition function to all orders in a perturbative expansion in the cutoff.

Both the canonical and path integral approach use widely different techniques to compute the finite cutoff partition function, yet, as expected from the equivalence between the two quantisation procedures, the results agree. Moreover, we find a perfect agreement between the result obtained via two approaches with a proposed deformation of the Schwarzian partition function, analogous to the $T\bar{T}$ deformation for 2D CFTs [9, 10]. Before diving in the computations, we first review (for completeness and later reference) this one-dimensional analog of $T\bar{T}$ and then present a more detailed summary of our results.

## 1.1 Review 1$d$ $T\bar{T}$

In previous work [9], a particular deformation of the Schwarzian quantum mechanics was shown to be classically equivalent to JT gravity with Dirichlet boundary conditions for the metric and dilaton. The deformation on the Schwarzian theory follows from a dimensional reduction of the $T\bar{T}$ deformation in 2D CFTs [3]. Explicitly the deformation involves a flow of

---

[2]For further investigations of Schwarzian quantum mechanics and JT gravity, see [12, 14–27].

[3]This reduction is valid in the classical limit and should be seen as a motivation for the proposed deformation. It would be interesting to extend it to a precise statement using the methods of [31].

the action $S$ of the quantum mechanical theory,

$$\partial_\lambda S = \int_0^1 d\theta \, \frac{T^2}{1/2 - 2\lambda T} \tag{1}$$

where $T$ is the trace of the stress-'scalar' of the quantum mechanical theory and $\lambda$ is the deformation parameter. By going from the Lagrangian to the Hamiltonian formulation, we can write an equivalent flow for the Hamiltonian instead of $S$ and find the flow of the energy eigenvalues,[4]

$$\partial_\lambda H = \frac{H^2}{1/2 - 2\lambda H} \quad \Rightarrow \quad \mathcal{E}_\pm(\lambda) = \frac{1}{4\lambda}\left(1 \mp \sqrt{1 - 8\lambda E}\right). \tag{2}$$

Here $E$ are the energy levels of the undeformed theory and matching onto the original spectrum as $\lambda \to 0$ results in picking the minus sign for the branch of the root in (2). In section 4 we will see that the other branch of the root will also make its appearance. In the case of the Schwarzian theory, which has a partition function that can be exactly computed [13],[5]

$$Z(\beta) = \int_0^\infty dE \, \frac{\sinh(2\pi\sqrt{2CE})}{\sqrt{2C\pi^3}} e^{-\beta E} = \frac{e^{2C\pi^2/\beta}}{\beta^{3/2}}, \tag{3}$$

the deformed partition function is,

$$Z_\lambda(\beta) = \int_0^\infty dE \, \frac{\sinh(2\pi\sqrt{2CE})}{\sqrt{2C\pi^3}} e^{-\beta \mathcal{E}_+(\lambda)}. \tag{4}$$

Let us make two observations. First, the integral over $E$ runs over the full positive real axis and therefore will also include complex energies $\mathcal{E}_+(\lambda)$ when $\lambda > 0$, i.e. for $E > 1/8\lambda$ the deformed spectrum complexifies. This violates unitarity and needs to be dealt with. We will come back to this issue in section 4. Second, given that there is a closed from expression of the original Schwarzian partition function, one can wonder whether this is also the case for the deformed partition function. This turns out to be the case. For the moment let us assume $\lambda < 0$ so that there are no complex energies, then it was shown in [9] that the deformed partition function is given by an integral transform of the original one, analogous to the result of [32] in 2$D$. The integral transform reads,

$$Z_\lambda(\beta) = \frac{\beta}{\sqrt{-8\pi\lambda}} \int_0^\infty \frac{d\beta'}{\beta'^{3/2}} e^{\frac{(\beta - \beta')^2}{8\lambda\beta'}} Z(\beta'), \tag{5}$$

Plugging (3) into this expression and performing the integral over $\beta'$ yields,

$$Z_\lambda(\beta) = \frac{\beta e^{-\frac{\beta}{4\lambda}}}{\sqrt{-2\pi\lambda}(\beta^2 + 16C\pi^2\lambda)} K_2\left(-\frac{1}{4\lambda}\sqrt{\beta^2 + 16C\pi^2\lambda}\right). \tag{6}$$

with the associated density of states given by

$$\rho_\lambda(E) = \frac{1 - 4\lambda E}{\sqrt{2\pi^3 C}} \sinh\left(2\pi\sqrt{2CE(1 - 2\lambda E)}\right) \tag{7}$$

Although we have derived this formula assuming that $\lambda < 0$, we will simply analytically continue to $\lambda > 0$ to obtain the partition function of the deformed Schwarzian theory that describes JT gravity at finite cutoff. One might be worried that this would not yield the same as (4) and indeed there are a few subtleties involved in doing that analytic continuation as discussed in the end of section 3 and in section 4.

---

[4]Since the deformation is a function of the Hamiltonian, the eigenfunctions do not change under the flow.

[5]In the gravitational theory $C$ is equal to $\phi_r$, the renormalised boundary value of the dilaton. Furthermore, here we picked a convenient normalisation of the partition function.

## 1.2 Summary of results and outline

The purpose of this paper is give two independent bulk computation that reproduce the partition function (4). In section 2 we present a derivation of the partition function of JT gravity (with negative cosmological constant) at finite cutoff by computing the radial Wheeler-de Witt (WdW) wavefunctional. Due to Henneaux it is known since the 80's that the contraints of $2D$ dilaton gravity can be solved exactly in the full quantum theory [29]. We will review this computation and fix the solution by imposing Hartle-Hawking boundary conditions. In particular we find that

$$\Psi_{\text{HH}}[\phi_b(u), L] = \int_0^\infty dM \ \sinh(2\pi\sqrt{M}) \ e^{\int_0^L du\left[\sqrt{\phi_b^2 - M - (\partial_u\phi_b)^2} - \partial_u\phi \tan^{-1}\left(\sqrt{\frac{\phi_b^2 - M}{(\partial_u\phi_b)^2} - 1}\right)\right]}. \tag{8}$$

This wavefunction is computed in a basis of fixed dilaton $\phi_b(u)$, where $u$ corresponds to the proper length along the boundary, and $L$ the total proper length of the boundary. The above results obtained through the WdW constraint are non-perturbative in both $L$ and $\phi_b(u)$.

When considering a constant dilaton profile $\phi_b(u) = \phi_b$, the wavefunction (8) reproduces the $T\bar{T}$ partition function in (4), with the identification

$$M \to 2CE, \quad \phi_b^2 \to \frac{C}{4\lambda}, \quad L \to \frac{\beta}{\sqrt{4C\lambda}}, \tag{9}$$

In terms of these variables, (8) matches with $T\bar{T}$ up to a shift in the ground state energy, which can be accounted for by a boundary counterterm $e^{-I_{\text{ct}}} = e^{-\phi_b L}$ added to the gravitational theory. An important aspect that this analysis emphasizes if the fact that, for JT gravity, studying boundary conditions with a constant dilaton is enough. As we explain in section 2.2, if the wavefunction for a constant dilaton is known, the general answer (8) is fixed by the constraints and does not constrain any further dynamical information [33].

The partition function (4) is also directly computed from the path integral in JT gravity at finite cutoff in section 3. We will impose dilaton and metric Dirichlet boundary conditions, in terms of $\phi_b$ and the total proper length $L$. For the reasons explain in the previous paragraph, it is enough to focus on the case of a constant dilaton. It is convenient to parametrize these quantities in the following way

$$\phi_b = \frac{\phi_r}{\varepsilon}, \quad L = \frac{\beta}{\varepsilon}, \tag{10}$$

in terms of a renormalized length $\beta$ and dilaton $\phi_r$. We will refer to $\varepsilon$ as the cutoff parameter [6]. When comparing with the $T\bar{T}$ approach this parameter is $\varepsilon = \sqrt{2\lambda}$ (in units for which we set $\phi_r \to 1/2$). In order to compare to the asymptotically $AdS_2$ case previously studied in the literature [11,23], we need to take $\phi_b, L \to \infty$ with a fixed renormalized length $L/\phi_b$. In terms of the cutoff parameter, this limit corresponds to $\varepsilon \to 0$, keeping $\phi_r$ and $\beta$ fixed.

We will solve this path integral perturbatively in the cutoff $\varepsilon$, to all orders. We integrate out the dilaton and reduce the path integral to a boundary action comprised of the extrinsic curvature $K$ and possible counter-terms. We find an explicit form of the extrinsic curvature valid to all orders in perturbation theory in $\varepsilon$. A key observation in obtaining this result is the realisation of a (local) $SL(2,\mathbb{R})$ invariance of $K$ in terms of lightcone coordinates $z = \tau - ix$, $\bar{z} = \tau + ix$:

$$K[z, \bar{z}] = K\left[\frac{az+b}{cz+d}, \frac{a\bar{z}+b}{c\bar{z}+d}\right]. \tag{11}$$

---

[6]In Poincaré coordinates $\varepsilon$ corresponds semiclassically to the bulk coordinate of the cutoff surface.

Solving the Dirichlet boundary condition for the metric allows us to write $K$ as a functional of the Schwarzian derivative of the coordinate $z$.[7] As we will explain in detail, the remaining path integral can be computed exactly using integrability properties in the Schwarzian theory to all orders in $\varepsilon^2$.

Thus, by the end of section 3, we find agreement between the WdW wavefunctional, the Euclidean partition function and the $T\bar{T}$ partition function from (4):

$$e^{-I_{\text{ct}}}\Psi_{\text{HH}}[\phi_b, L] \overset{\text{non-pert.}}{=} Z_\lambda(\beta) \overset{\text{pert.}}{=} Z_{\text{JT}}[\phi_b, L]. \tag{12}$$

Here we emphasize again that we show that the first equality is true non-perturbatively in $\varepsilon$ (respectively in $\lambda$), whereas we prove the second equality to all orders in perturbation theory.

In section 4 we discuss various extensions of the deformed partition function including further corrections. In particular we discuss two types of corrections in the path integral and in the integral over energies in (4): first, we analyze non-perturbative terms in $\varepsilon$ coming from contributions that cannot be written as a path integral on the disk (the contracting branch of the wavefunction) and second, we speculate about non-perturbative corrections coming from the genus expansion. Related to the first kind of ambiguity, given the exact results we obtained for the wavefunctional and partition function, we explore how the complexification of the energy levels (that we mentioned above) can be cured. In particular, we propose that it requires the inclusion of the other branch of the root in (2), but still results in a negative density of states. The structure of the negative density of states suggests that the (unitary) partition function is not an ordinary one, but one with a chemical potential turned on. Related to the second type, we compute the partition function of the finite cutoff "trumpet" which is a necessary ingredient when constructing higher genus hyperbolic surfaces. Finally, we speculate about the range of the remaining Weil-Petersson integral which is needed in order to compute the finite cutoff partition function when including the contribution of surfaces with arbitrary topology.

Section 5 applies the computation from section 2 to the case of JT gravity with a positive cosmological constant and finds the wavefunctional on a de Sitter time-slice at finite time. This wavefunctional has some interesting behaviour, similar to the Hagedorn divergence present in (6). We finish with a discussion of our results and future directions in section 6.

**Note:** While this work was in progress, we became aware of a closely related project by D. Stanford and Z. Yang [34]. They analyze finite cutoff JT gravity from yet a different perspective, finding different results. We leave understanding how these approaches are related for future work, but we believe they correspond to different ways to regularize (and therefore define) the theory.

## 2 Wheeler-DeWitt wavefunction

In this section, we will start by reviewing the canonical quantization of 2$D$ dilaton-gravity following the approach of [29, 30]. In these references, the authors find the space of exact solutions for both the momentum and Wheeler-DeWitt constraints. Later, in subsections 2.3 and 2.4, we will focus on JT gravity, and we will explain how to impose the Hartle-Hawking condition appropriately to pick a solution corresponding to finite cutoff AdS$_2$.

Let us consider the more general two dimensional dilaton gravity in Lorentzian signature,

$$I = \frac{1}{2}\int_M d^2x\sqrt{g}[\phi R - U(\phi)] + \int_{\partial M} du\sqrt{\gamma_{uu}}\,\phi K, \tag{13}$$

---

[7]This generalizes the computation of [11] which found the relation between the extrinsic curvature and the Schwarzian derivative in the infinite cutoff limit.

with an arbitrary potential $U(\phi)$. $g$ is the two-dimensional space-time metric on $M$ and $\gamma$ the induced metric on its boundary $\partial M$. The boundary term in (13) is necessary in order for the variational principle to be satisfied when imposing Dirichlet boundary conditions for the metric and dilaton. In (13) we could also add the topological term

$$\frac{1}{2}\int_M d^2x \sqrt{g}\, \phi_0 R + \int_{\partial M} du \sqrt{\gamma_{uu}}\, \phi_0 K = 2\pi \phi_0, \tag{14}$$

which will be relevant in section 4.3.

It will be useful to define also the prepotential $W(\phi)$ by the relation $\partial_\phi W(\phi) = U(\phi)$. In the case of JT gravity with negative (or positive) cosmological constant we will pick $U(\phi) = -2\phi$ (or $U(\phi) = 2\phi$) and $W(\phi) = -\phi^2$ ($W(\phi) = \phi^2$), which has as a metric solution AdS$_2$ (dS$_2$) space with unit radius.

We will assume the topology of space to be a closed circle, and will use the following ADM decomposition of the metric

$$ds^2 = -N^2 dt^2 + h(dx + N_\perp dt)^2, \quad h = e^{2\sigma} \tag{15}$$

where $N$ is the lapse, $N_\perp$ the shift, $h$ the boundary metric (which in this simple case is an arbitrary function of $x$) and we identify $x \sim x + 1$. After integrating by parts and using the boundary terms, the action can then be written as

$$\begin{aligned} I \;=\; & \int d^2x \, e^\sigma \Big[ \frac{\dot{\phi}}{N}(N_\perp \partial_x \sigma + \partial_x N_\perp - \dot{\sigma}) \\ & + \frac{\partial_x \phi}{N}\left( \frac{N \partial_x N}{e^{2\sigma}} - N_\perp \partial_x N_\perp + N_\perp \dot{\sigma} - N_\perp^2 \partial_x \sigma \right) - \frac{1}{2} N U(\phi) \Big] \end{aligned} \tag{16}$$

where the dots correspond to derivatives with respect to $t$. As usual the action does not involve time derivatives of fields $N$ and $N_\perp$ and therefore

$$\Pi_N = \Pi_{N_\perp} = 0, \tag{17}$$

which act as primary constraints. The momenta conjugate to the dilaton and scale factor are

$$\Pi_\phi = \frac{e^\sigma}{N}(N_\perp \partial_x \sigma + \partial_x N_\perp - \dot{\sigma}), \quad \Pi_\sigma = \frac{e^\sigma}{N}(N_\perp \partial_x \phi - \dot{\phi}). \tag{18}$$

With these equations we can identify the momentum conjugate to the dilaton with the extrinsic curvature $\Pi_\phi \sim K$, and the momentum of $\sigma$ with the normal derivative of the dilaton $\Pi_\sigma \sim \partial_n \phi$. The classical Hamiltonian then becomes

$$H = \int dx \left[ N_\perp \mathcal{P} + e^{-\sigma} N \mathcal{H}_{\text{WdW}} \right] \tag{19}$$

where

$$\mathcal{P} \;\equiv\; \Pi_\sigma \partial_x \sigma + \Pi_\phi \partial_x \phi - \partial_x \Pi_\sigma, \tag{20}$$

$$\mathcal{H}_{\text{WdW}} \;\equiv\; -\Pi_\phi \Pi_\sigma + \frac{1}{2} e^{2\sigma} U(\phi) + \partial_x^2 \phi - \partial_x \phi \partial_x \sigma, \tag{21}$$

and classically the momentum and Wheeler-DeWitt constraints are respectively $\mathcal{P} = 0$ and $\mathcal{H}_{\text{WdW}} = 0$.

So far the discussion has been classical. Now we turn to quantum mechanics by promoting field to operators. We will be interested in wavefunctions obtained from path integrals over the

metric and dilaton, and we will write them in configuration space. The state will be described by a wave functional $\Psi[\phi,\sigma]$ and the momentum operators are replaced by

$$\hat{\Pi}_\sigma = -i\frac{\delta}{\delta\sigma(x)}, \quad \hat{\Pi}_\phi = -i\frac{\delta}{\delta\phi(x)}, \tag{22}$$

The physical wavefunctions will only depend on the boundary dilaton profile and metric.

Usually, when quantizing a theory, one needs to be careful with the measure and whether it can contribute Liouville terms to the action. Such terms only appear when in conformal gauge, which is not what we are working in presently. Actually, the ADM decomposition (15) captures a general metric and is merely a parametrization of all $2D$ metrics and so we have not fixed any gauge. The quantum theory is thus defined through the quantum mechanical version of the classical constraints (20) and (21) [8]. As a result, we do not need to include any Liouville term in our action in the case of pure gravity. If matter would have been present, there could be Liouville terms coming from integrating out the matter, but that is beyond the scope of this paper.

## 2.1 Solution

In references [29,30], the physical wavefunctions that solve the dilaton gravity constraints are constructed as follows. The key step is to notice that the constraints $\mathcal{P}$ and $\mathcal{H}_{\mathrm{WdW}}$ are simple enough that we can solve for $\Pi_\sigma$ and $\Pi_\phi$ separately. For instance, by combing $\Pi_\sigma\mathcal{P}$ with the WdW constraint, we get

$$\partial_x(e^{-2\sigma}\Pi_\sigma^2) = \partial_x(e^{-2\sigma}(\partial_x\phi)^2 + W(\phi)) \Rightarrow \Pi_\sigma = \pm\sqrt{(\partial_x\phi)^2 + e^{2\sigma}[M + W(\phi)]}, \tag{23}$$

with $M$ an integration constant that is proportional to the ADM mass of the system as we will see momentarily. It is then straightforward to plug this into the WdW constraint to find an expression for $\Pi_\phi$. Quantum mechanically, we want the physical wavefunction to satisfy,

$$\hat{\Pi}_\sigma\Psi_{\mathrm{phys}} = \pm Q[M;\phi,\sigma]\Psi_{\mathrm{phys}}, \quad \hat{\Pi}_\phi\Psi_{\mathrm{phys}} = \pm\frac{g[\phi,\sigma]}{Q[M;\phi,\sigma]}\Psi_{\mathrm{phys}}, \tag{24}$$

where we defined the functions

$$Q[E;\phi,\sigma] \equiv \sqrt{(\partial_x\phi)^2 + e^{2\sigma}[M + W(\phi)]}, \quad g[\phi,\sigma] \equiv \frac{1}{2}e^{2\sigma}U(\phi) + \partial_x^2\phi - \partial_x\phi\partial_x\sigma. \tag{25}$$

Wavefunctions that solve these constraints also solve the momentum and Wheeler-DeWitt constraints as explained in [29, 30]. In particular they solve the following WdW equation with factor ordering,[9]

$$\left(g - \hat{Q}\hat{\Pi}_\phi\hat{Q}^{-1}\hat{\Pi}_\sigma\right)\Psi_{\mathrm{phys}} = 0 \tag{26}$$

The most general solution can be written as

$$\Psi = \Psi_+ + \Psi_-, \quad \Psi_\pm = \int dM\rho_\pm(M)\Psi_\pm(M), \tag{27}$$

---

[8]From the path integral perspective, we are assuming an infinite range of integration over the lapse. Different choices for the contour of integration can drastically modify the constraints after quantization. We thank S. Giddings for discussions on this point.

[9]Here we think of $\hat{Q}$ as well as $\hat{M}$ as operators. The physical wavefunctions can be written as linear combinations of eigenfunctions of the operator $\hat{M}$ with eigenvalue $M$.

where we will distinguish the two contributions

$$\Psi_\pm(M) = \exp\left[\pm i \int dx \left(Q[M;\phi,\sigma] - \partial_x\phi \tanh^{-1}\left(\frac{Q[M;\phi,\sigma]}{2\partial_x\phi}\right)\right)\right], \tag{28}$$

with the function $Q$ defined in (25) which depends on the particular dilaton potential. We will refer in general to $\Psi_+$ ($\Psi_-$) as the expanding (contracting) branch.

This makes explicit the fact that solutions to the physical constraints reduce the naive Hilbert space from infinite dimensional to two dimensional with coordinate $M$ (and its conjugate). The most general solution of the Wheeler-DeWitt equation can then be expanded in the base $\Psi_\pm(M)$ with coefficients $\rho_\pm(M)$. The new ingredient in this paper will be to specify appropriate boundary conditions to pick $\rho_\pm(M)$ and extract the full Hartle-Hawking wavefunction. We will see this is only possible for JT gravity for reasons that will be clear in the next section.

It will be useful to write the physical wavefunction in terms of diffeomorphism invariant quantities. This is possible thanks to the fact that we are satisfying the momentum constraints. In order to do this we will define the proper length $u$ of the spacelike circle as

$$du = e^\sigma dx, \quad L \equiv \int_0^1 e^\sigma dx, \tag{29}$$

where $L$ denotes the total length. The only gauge invariant data that the wavefunction can depend on is then $L$ and $\phi(u)$, a dilaton profile specified as a function of proper length along the boundary. The wavefunction (28) can be rewritten as

$$\Psi_\pm(M) = e^{\pm i \int_0^L du \left[\sqrt{W(\phi)+M+(\partial_u\phi)^2} - \partial_u\phi \tanh^{-1}\left(\sqrt{1+\frac{W(\phi)+M}{(\partial_u\phi)^2}}\right)\right]}, \tag{30}$$

which is then manifestly diffeomorphism invariant.

The results of this section indicate the space of physical states that solve the gravitational constraints is one dimensional, labeled by $M$. In the context of radial quantization of $AdS_2$ that we will analyze in the next section, this parameter corresponds to the ADM mass of the state, while in the case of $dS_2$, it corresponds to the generator of rotations in the spatial circle. Phase space is even-dimensional, and the conjugate variable to $M$ is given by

$$\Pi_M = -\int dx \frac{e^{2\sigma}\Pi_\rho}{\Pi_\rho^2 - 2(\partial_x\phi)^2} \tag{31}$$

such that $[M, \Pi_M] = i$.[10]

## 2.2 Phase space reduction

Having the full solution to the WdW equation, we now study the minisuperspace limit. In this limit, the dilaton $\phi$ and boundary metric $e^{2\sigma}$ are taken to be constants. In a general theory of gravity, minisuperspace is an approximation. In JT gravity, as we saw above, the physical phase space is finite-dimensional (two dimensional to be precise). Therefore giving the wavefunction in the minisuperspace regime encodes all the dynamical information of the theory, while the generalization to varying dilaton is fixed purely by the constraints. In this section, we will directly extract the equation satisfied by the wavefunction as a function of constant dilaton and metric, from the more general case considered in the previous section.

---

[10]The simplicity of the phase space of dilaton gravity theories was also noted in [35].

If we start with the WdW equation and fix the dilaton and metric to be constant, the functional derivatives then become ordinary derivatives and the equation reduces to

$$\left(\frac{1}{2}e^{2\sigma}U(\phi) - \hat{Q}\partial_\phi\hat{Q}^{-1}\partial_\sigma\right)\Psi(\phi,\sigma) = 0. \tag{32}$$

with $\hat{Q} = (\hat{M} + W(\phi))^{1/2}$. Due to the factor ordering, this differential equation still depends on the operator $\hat{M}$, which is a bit unsatisfactory. Fortunately, we know that a $\sigma$ derivative acting on $\Psi$ is the same as acting with $Q^2/g\partial_\phi$. In the minisuperspace limit, we can therefore write (32) as

$$(LU(\phi) - 2L\partial_L(L^{-1}\partial_\phi))\Psi(\phi,L) = 0, \tag{33}$$

where $L$ is the total boundary length. This equation is the exact constraint that wavefunctions with a constant dilaton should satisfy even though it was derived in a limit. We can explicitly check this by using (28) and noticing that any physical wavefunction, evaluated in the minisuperspace limit, will satisfy precisely this equation.

This equation differs from the one obtained in [33] by $\Psi_{\text{here}} = L\Psi_{\text{there}}$ and, therefore, changes the asymptotics of the wavefunctions, something we will analyze more closely in the next subsection.

## 2.3 Wheeler-DeWitt in JT gravity: radial quantization

In this section, we will specialize the previous discussion to JT gravity with a negative cosmological constant. We fix units such that $U(\phi) = -2\phi$. We will analytically continue the results of the previous section to Euclidean space and interpret them in the context of radial quantization, such that the wavefunction is identified with the path integral in a finite cutoff surface. Then, we will explain how to implement Hartle-Hawking boundary conditions, obtaining a proposal for the exact finite cutoff JT gravity path integral that can be compared with results for the analog of the $T\bar{T}$ deformation in $1d$ [9,10].

Lets begin by recalling some small changes that appear when going from Lorenzian to Euclidean radial quantization. The action we will work with is

$$I_{\text{JT}} = -\frac{1}{2}\int_M \sqrt{g}\phi(R+2) - \int_{\partial M}\sqrt{\gamma}\phi K, \tag{34}$$

and the ADM decomposition of the metric we will use is

$$ds^2 = N^2 dr^2 + h(d\theta + N_\perp dr)^2, \quad h = e^{2\sigma}, \tag{35}$$

where $r$ is the radial direction while $\theta \sim \theta + 1$ corresponds to the angular direction that we will interpret as Euclidean time. We show these coordinates in figure 1. In terms of holography we will eventually interpret $\theta$ as related to the Euclidean time of a boundary quantum mechanical theory.

As shown in figure 1, and as we will explicitly show in section 3, the radial quantization wavefunction is identified with the gravitational path integral at a finite cutoff (inside the black circle) with Dirichlet boundary conditions

$$\Psi[\phi_b(u), \sigma(u)] = \int \mathcal{D}g\mathcal{D}\phi \ e^{-I_{\text{JT}}[\phi,g]}, \quad \text{with} \quad \phi|_\partial = \phi_b(u), \quad g|_\partial = \gamma_{uu} = e^{2\sigma(u)}. \tag{36}$$

The geometry inside the disk in figure 1 is asymptotically $EAdS_2$. From this path integral we can derive the WDW and momentum constraints and therefore solving the latter with the appropriate choice of state should be equivalent to doing the path integral directly.

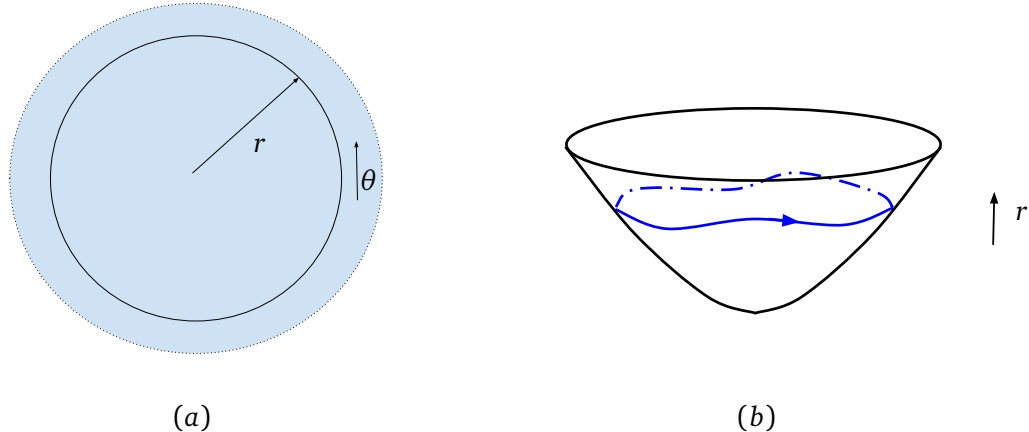

$$(a) \hspace{5cm} (b)$$

Figure 1: (a) We show the slicing we use for Euclidean JT gravity in asymptotically $AdS_2$, which has disk topology (but not necessarily rigid hyperbolic metric). (b) Frame where the geometry is rigid $EAdS_2$ with $r$ increasing upwards and a wiggly boundary denoted by the blue curve.

The result of previous section implies that this path integral is given by a linear combination of

$$\text{Expanding branch:} \quad \Psi_+(M) = e^{\int_0^L du \left[ \sqrt{\phi_b^2 - M - (\partial_u \phi_b)^2} - \partial_u \phi_b \tan^{-1}\left( \sqrt{\frac{\phi_b^2 - M}{(\partial_u \phi_b)^2} - 1} \right) \right]}, \tag{37}$$

$$\text{Contracting branch:} \quad \Psi_-(M) = e^{-\int_0^L du \left[ \sqrt{\phi_b^2 - M - (\partial_u \phi_b)^2} - \partial_u \phi_b \tan^{-1}\left( \sqrt{\frac{\phi_b^2 - M}{(\partial_u \phi_b)^2} - 1} \right) \right]}. \tag{38}$$

We will focus on the purely expanding branch of the solution (37), as proposed in [36] and [33] to correspond to the path integral in the disk and therefore set $\rho_-(M) = 0$. We will go back to possible effects coming from turning on this term later. Thus, we will study the solutions

$$\Psi_{\text{disk}}[\phi_b(u), \sigma(u)] = \int dM \rho(M) \, e^{\int_0^L du \left[ \sqrt{\phi_b^2 - M - (\partial_u \phi_b)^2} - \partial_u \phi_b \tan^{-1}\left( \sqrt{\frac{\phi_b^2 - M}{(\partial_u \phi_b)^2} - 1} \right) \right]}. \tag{39}$$

To make a choice of boundary conditions that fix the boundary curve very close to the boundary of the disk we will eventually take the limit of large $L$ and $\phi_b$.

## 2.4 Hartle-Hawking boundary conditions and the JT wavefunctional

To determine the unknown function $\rho(M)$, we will need to impose a condition that picks the Hartle-Hawking state. For this, one usually analyses the limit $L \to 0$ [37]. Such a regime is useful semiclassically but not in general. From the no-boundary condition, $L \to 0$ should reproduce the path integral over JT gravity inside tiny patches deep inside the hyperbolic disk; performing such a calculation is difficult. Instead, it will be simpler to impose the Hartle-Hawking condition at large $L \to \infty$. In this case, we know how to do the path integral directly using the Schwarzian theory. The derivation of the Schwarzian action from [11] explicitly uses the no-boundary condition, so we will take this limit instead, which will be enough to identify a preferred solution of the WdW equation.

To match the wavefunction with the partition function of the Schwarzian theory, it is enough to consider the case of constant dilaton and metric. Then, the wavefunction simplifies

to[11]

$$\Psi[\phi_b, \sigma] = \int dM \rho(M) \, e^{\int_0^1 d\theta e^\sigma \sqrt{\phi_b^2 - M}} = \int dM \rho(M) \, e^{\int_0^L du \sqrt{\phi_b^2 - M}} \tag{40}$$

with $\phi_b$ and $\sigma$ constants. Expanding the root at large $\phi_b$ and large $L = e^\sigma$ gives,

$$\Psi[\phi_b, \sigma] = e^{L\phi_b} \int dM \rho(M) \, e^{-L \frac{M}{2\phi_b} + \cdots} \tag{41}$$

We find the usual divergence for large $L$ and $\phi$, which can be removed by adding to (34) the counter term, $I_{\text{ct}} = \int_0^L du \, \phi_b$. In fact, we will identify the JT path integral with this counter term as computing the thermal partition function at a temperature specified by the boundary conditions. At large $L$ and $\phi_b$ we know that the gravity partition function is given by the Schwarzian theory:

$$\int \mathcal{D}g \mathcal{D}\phi \, e^{-I_{\text{JT}}[\phi,g]} \rightarrow e^{L\phi_b} \int \frac{\mathcal{D}f}{SL(2,\mathbb{R})} \, e^{\phi_b \int_0^L du \, \text{Sch}(\tan \frac{\pi}{L} f, u)}, \tag{42}$$

where $\text{Sch}(F(u), u) \equiv \frac{F'''}{F'} - \frac{3}{2} \left( \frac{F''}{F'} \right)^2$. By rescaling time we can see the path integral only depends on $L/\phi_b$ which we will sometimes refer to as renormalized length. This result can be derived by first integrating over the dilaton over an imaginary contour, localizing the geometry to rigid $AdS_2$. Then the remaining degree of freedom is the shape of the boundary curve, from which the Schwarzian theory arises.

The Schwarzian partition function can be computed exactly and gives

$$Z_{\text{Sch}}(\ell) \equiv \int \frac{\mathcal{D}f}{SL(2,\mathbb{R})} e^{\int_0^\ell du \, \text{Sch}(\tan \frac{\pi}{\ell} f, u)} = \left( \frac{\pi}{\ell} \right)^{3/2} e^{\frac{2\pi^2}{\ell}} = \int dk^2 \sinh(2\pi k) e^{-\ell k^2/2} \tag{43}$$

Applying this result to the JT gravity path integral with the replacement $\ell \rightarrow L/\phi_b$ gives the partition function directly in the form of equation (41) where we can straightforwardly identify the Schwarzian density of states with the function of $M$ as

$$\rho_{\text{HH}}(M) = \sinh(2\pi \sqrt{M}), \tag{44}$$

where the subscript indicates that we picked the Hartle-Hawking state. It is important that we are able to compute the path integral of JT gravity for $\phi_b, L \rightarrow \infty$ but fixed $L/\phi_b$. This involves an exact treatment of the Schwarzian mode since otherwise we would only obtain $\rho_{\text{HH}}(M)$ in some limits. This ingredient was missing in [29,30] making them unable to identify the HH state from the full space of physical states.

To summarize, the solution of the gravitational constraints gives the finite cutoff JT gravity path integral as

$$\Psi_{\text{HH}}[\phi_b(u), L] = \int_0^\infty dM \sinh(2\pi \sqrt{M}) \, e^{\int_0^L du \left[ \sqrt{\phi_b^2 - M - (\partial_u \phi_b)^2} - \partial_u \phi \tan^{-1} \left( \sqrt{\frac{\phi^2 - M}{(\partial_u \phi)^2} - 1} \right) \right]}. \tag{45}$$

By construction, this matches the Schwarzian limit when $\phi_b$ and $\sigma$ are constant.

When the dilaton is constant but $\sigma(u)$ is not, it is clear that we can simply go to coordinates $d\tilde{\theta} = e^\sigma d\theta$ in both the bulk path integral and the WdW wavefunction and see that they give the same result. Since we can always choose time-slices with a constant value for the dilaton, this situation will suffice for comparing our result to the analog of the $T\bar{T}$ deformation in the next subsection.

The more non-trivial case is for non-constant dilaton profiles. We provide a further check of our result in appendix A.1, where we compare the wavefunctional (45) to the partition function of JT gravity with a non-constant dilaton profile when the cutoff is taken to infinity.

---

[11]It is interesting to note that this partition function first appeared in [38].

## 2.5 Comparison to $T\bar{T}$

Let us now compare the wavefunctional (45) to the partition function obtained from the $1D$ analog of the $T\bar{T}$ deformation (4). First of all, let us consider configurations of constant $\phi_b$, so $\partial_u \phi_b = 0$. This will simplify $\Psi_{\text{HH}}$ to

$$\Psi_{\text{HH}}[\phi_b, L] = \int_0^\infty dM \ \sinh(2\pi\sqrt{M})e^{\phi_b L\sqrt{1-M/\phi_b^2}}. \tag{46}$$

The partition function is then obtained by multiplying this wavefunction by $e^{-I_{\text{ct}}} = e^{-L\phi_b}$. The resulting partition function agrees with (4) with identifications:

$$M \to 2CE, \quad \phi_b^2 \to \frac{C}{4\lambda}, \quad L \to \frac{\beta}{\sqrt{4C\lambda}}, \tag{47}$$

up to an unimportant normalization. In fact, we can say a little more than just mapping solutions onto each other. In section 2.2 we showed that in the minisuperspace approximation the wave*functions* satisfy (33). With the identifications made above and the inclusion of the counter term, the partition function $Z_\lambda(\beta)$ satisfies

$$\left[4\lambda\partial_\lambda\partial_\beta + 2\beta\partial_\beta^2 - \left(\frac{4\lambda}{\beta} - 1\right)\partial_\lambda\right]Z_\lambda(\beta) = 0. \tag{48}$$

This is now purely written in terms of field theory variables and is precisely the flow equation as expected from (1), i.e. solutions to this differential equation have the deformed spectrum (2). This is also the flow of the partition function found in two dimensions in [39], specialised to purely imaginary modular parameter of the torus. We will analyze the associated non-perturbative ambiguities associated to this flow in section 4.

Let us summarise. We have seen that the partition function of the deformed Schwarzian theory is mapped to the exact dilaton gravity wavefunctions for constant $\phi_b$ and $\gamma_{uu}$. In fact, *any* quantum mechanics theory that is deformed according to (1) will obey the quantum WdW equation (for constant $\phi_b$ and $\sigma$). This principle can be thought of as the two-dimensional version of [36]. It is only the boundary condition at $\lambda \to 0$ (or large $\phi_b L$), where we know the bulk JT path integral gives the Schwarzian theory, that tells us that the density of states is $\sinh(2\pi\sqrt{M})$. Next, we will show that the wavefunction for constant $\phi_b$ and $\gamma_{uu}$ can be reproduced by explicitly computing the Euclidean path integral in the bulk, at finite cutoff.

## 3 The Euclidean path integral

We will once again consider the JT gravity action, (34), and impose Dirichlet boundary conditions for the dilaton field $\phi|_{\partial M_2} \equiv \phi_b \equiv \phi_r/\varepsilon$, boundary metric $\gamma_{uu}$, and proper length $L \equiv \beta/\varepsilon$ and with the addition the counter-term,

$$I_{\text{ct}} = \int du\sqrt{\gamma}\phi, \tag{49}$$

whose addition leads to an easy comparison between our results and the infinite cutoff results in JT gravity. As in the previous section we will once again focus on disk topologies.

As discussed in section 2.4, the path integral over the dilaton $\phi$ yields a constraint on the curvature of the space, with $R = -2$. Therefore, in the path integral we are simply summing over different patches of $AdS_2$, which we parametrize in Euclidean signature using Poincaré coordinates as $ds^2 = (d\tau^2 + dx^2)/x^2$. To describe the properties which we require of the

boundary of this patch we choose a proper boundary time $u$, with a fixed boundary metric $\gamma_{uu} = 1/\varepsilon^2$ (related to the fixed proper length $L = \int_0^\beta du \sqrt{\gamma_{uu}}$). Fixing the intrinsic boundary metric to a constant, requires:

$$\frac{\tau'^2 + x'^2}{x^2} = \frac{1}{\varepsilon^2}, \qquad \frac{-t'^2 + x'^2}{x^2} = \frac{1}{\varepsilon^2}, \qquad \tau = -it. \tag{50}$$

If choosing some constant $\varepsilon \in \mathbb{R}$ then we require that the boundary has the following properties:

- If working in Euclidean signature, the boundary should never self-intersect. Consequently if working on manifolds with the topology of a disk this implies that the Euler number $\chi(M_2) = 1$.

- If working in Lorentzian signature, the boundary should always remain time-like since (50) implies that $-(t')^2 + (x')^2 = (x' - t')(t' + x') > 0$.[12] From now on we will assume without loss of generality that $t' > 0$.

Both conditions are important constraints which we should impose at the level of the path integral. Such conditions are not typical if considering the boundary of the gravitational theory as the worldline of a particle moving on $H^2$ or $AdS_2$: in Euclidean signature, the worldline could self-intersect, while in Lorentzian signature the worldline could still self-intersect but could also become space-like. These are the two deficiencies that [21, 22] encountered in their analysis, when viewing the path integral of JT gravity as that of a particle moving in an imaginary magnetic field on $H^2$.

For the purposes of this paper it will also prove convenient to introduce the light-cone coordinates (with $z = -ix + \tau, \bar{z} = ix + \tau$), for which fixing the intrinsic boundary metric implies:

$$-\frac{4z'\bar{z}'}{(z - \bar{z})^2} = \frac{1}{\varepsilon^2}. \tag{51}$$

In Euclidean signature $z = \bar{z}^*$, while in Lorentzian signature $z, \bar{z} \in i\mathbb{R}$. The constraint that the boundary is time-like implies that $iz' > 0$ and $i\bar{z}' < 0$ (alternatively, if assuming $t' < 0$, $iz' < 0$ and $i\bar{z}' > 0$). In order to solve the path integral for the remaining boundary fluctuations in the 1D system it will prove convenient to use light-cone coordinates and require that the path integral obeys the two properties described above.

## 3.1 Light-cone coordinates and $SL(2, \mathbb{R})$ isometries in $AdS_2$

As is well known, $AdS_2$, even at finite cutoff, exhibits an $SL(2, \mathbb{R})$ isometry. This isometry becomes manifest when considering the coordinate transformations:

$$\text{E \& L:} \qquad z \to \frac{az + b}{cz + d}, \qquad\qquad\qquad \bar{z} \to \frac{a\bar{z} + b}{c\bar{z} + d},$$

$$\text{E:} \qquad x + i\tau \to \frac{a(x + i\tau) + b}{c(x + i\tau) + d}, \qquad \text{L:} \quad t + x \to \frac{a(t + x) + b}{c(t + x) + d}, \tag{52}$$

It is straightforward to check that under such transformations the boundary metrics, (50) and (51), both remain invariant. The same is true of the extrinsic curvature, which is the light-cone parametrization of the boundary degrees of freedom can be expressed as

$$K[z(u), \bar{z}(u)] = \frac{2z'^2\bar{z}' + (\bar{z} - z)\bar{z}'z'' + z'(2\bar{z}'^2 + (z - \bar{z})\bar{z}'')}{4(z'\bar{z}')^{3/2}}. \tag{53}$$

---

[12]While fixing the metric $\gamma_{uu}$ to be a constant is not diffeomorphism invariant, the notion of the boundary being time-like (sgn $\gamma_{uu}$) is in fact diffeomorphism invariant.

Consequently, invariance under $SL(2, \mathbb{R})$ transformations gives:

$$K[z, \bar{z}] = K\left[\frac{az + b}{cz + d}, \frac{a\bar{z} + b}{c\bar{z} + d}\right], \tag{54}$$

Therefore, upon solving for $\bar{z}[z(u)]$ (as a functional of $z(u)$) we will find that

$$\bar{z}[z(u)] \qquad \Rightarrow \qquad K[z] = K\left[\frac{az + b}{cz + d}\right] \tag{55}$$

As we will see in the next subsection, such a simple invariance under $SL(2, \mathbb{R})$ transformations will be crucial to being able to relate the path integral of the boundary fluctuations to that of some deformation of the Schwarzian theory. An important related point is that when solving for $\tau[x(u)]$ as a functional of $x(u)$, the resulting extrinsic curvature is not invariant under the $SL(2, \mathbb{R})$ transformations, $\tau \to \frac{a\tau + b}{c\tau + d}$. Rather this is only a valid symmetry in the $\varepsilon \to 0$ limit, for which $x \to 0$, while $\tau$ is kept finite. It is only in the asymptotically $AdS_2$ limit that the transformation in the second line of (52) can be identified with $\tau \to \frac{a\tau + b}{c\tau + d}$. If keeping track of higher orders in $\varepsilon$, the transformation on $\tau$ would involve a growing number of derivatives on the $\tau$ field which should be proportional to the order of the $\varepsilon$-expansion.

## 3.2 Restricting the extrinsic curvature

Next, we discuss the expansion of the extrinsic curvature $K[z]$ to all orders in perturbation in $\varepsilon$:

$$K[z] = \sum_{n=0}^{\infty} \varepsilon^n K_n[z], \qquad K_n[z] = K_n\left[\frac{az + b}{cz + d}\right], \tag{56}$$

We could in principle explicitly solve for $\bar{z}[z(u)]$ to first few orders in perturbation theory in $\varepsilon$ and then plug the result into (64). The first few orders in the expansion can be solved explicitly and yield:

$$K_0[z] = 1, \qquad K_1[z] = 0, \qquad K_2[z] = \text{Sch}(z, u),$$
$$K_3[z] = -i\, \partial_u \text{Sch}(z, u), \qquad K_4[z] = -\frac{1}{2}\text{Sch}(z, u)^2 + \partial_u^2 \text{Sch}(z, u). \tag{57}$$

The fact that all orders in $K_n[z(u)]$ solely depend on the Schwarzian and its derivatives is not a coincidence. In fact, one generally finds that:

$$K_n[z] = \mathcal{K}_n[\text{Sch}(z, u), \partial_u]. \tag{58}$$

The reason for this is as follows. $K_n[z]$ is a local function of $z(u)$ since solving for $\bar{z}[z(u)]$ involves only derivatives of $z(u)$. The Schwarzian can be written as the Casimir of the $\mathfrak{sl}(2, \mathbb{R})$ transformation, $z \to \frac{az + b}{cz + d}$ [11]. Because the rank of the $\mathfrak{sl}(2, \mathbb{R})$ algebra is 1, higher-order Casimirs of $\mathfrak{sl}(2, \mathbb{R})$ can all be expressed as a polynomial (or derivatives of powers) of the quadratic Casimir. Since local functions in $u$ that are $SL(2, \mathbb{R})$ invariant, can also only be written in terms of the Casimirs of $\mathfrak{sl}(2, \mathbb{R})$ this implies that they should also be linear combinations of powers (or derivatives of powers) of the quadratic Casimir, which is itself the Schwarzian.

Alternatively, we can prove that $K_n[z(u)]$ is a functional of the Schwarzian by once again noting that $K_n[z(u)]$ only contains derivatives of $z(u)$ up to some finite order. Then we can check explicitly how each infinitesimal $SL(2, \mathbb{R})$ transformation constrains $K_n[z(u)]$. For instance, translation transformations $z \to z + b$ imply that $K_n$ solely depends on derivatives of $z(u)$. The transformation $z(u) \to az(u)$ implies that $K_n[z(u)]$ depends solely on ratios of derivatives with a matching order in $z$ between the numerator and denominator of each ratio,

of the type $(\prod_k z^{(k_i)})/(\prod_k z^{(\tilde{k}_i)})$. Finally considering all possible linear combinations between ratios of derivatives of the type $(\prod_k z^{(k_i)})/(\prod_k z^{(\tilde{k}_i)})$ and requiring invariance under the transformation $z(u) \rightarrow 1/z(u)$, fixes the coefficients of the linear combination to those encountered in arbitrary products of Schwarzians and of its derivatives.

Once again, we emphasize that this does not happen when using the standard Poincaré parametrization (50) in $\tau$ and $x$. When solving for $\tau[x]$ and plugging into $K[\tau(u)]$, since we have that $K[\tau(u)] \neq K[a\tau(u) + b/(c\tau(u) + d)]$ and consequently $K[\tau(u)]$ is not a functional of the Schwarzian; it is only a functional of the Schwarzian at second-order in $\varepsilon$. This can be observed by going to fourth order in the $\varepsilon$-expansion, where

$$K_4[\tau(u)] = \frac{\tau^{(3)}(u)^2}{\tau'(u)^2} + \frac{27\tau''(u)^4}{8\tau'(u)^4} + \frac{\tau^{(4)}(u)\tau''(u)}{\tau'(u)^2} - \frac{11\tau^{(3)}(u)\tau''(u)^2}{2\tau'(u)^3}, \tag{59}$$

which cannot be written in terms of $\mathrm{Sch}(\tau(u), u)$ and of its derivatives.

### 3.3 Finding the extrinsic curvature: perturbative terms in $K[z(u)]$

The previous subsection identified the abstract dependence of the extrinsic curvature as a function of the Schwarzian. To quantize the theory, we need to find the explicit dependence of $K_n$ on the Schwarzian. To do this, we employ the following trick. Consider the specific configuration for $z(u)$:[13]

$$z(u) = \exp(au), \qquad \mathrm{Sch}(z, u) = -\frac{a^2}{2}. \tag{60}$$

Since $K[z(u)]$ is a functional of the $\mathrm{Sch}(z, u)$ and of its derivatives to all orders in perturbation theory in $\varepsilon$, then $K_n[z(u) = \exp(au)] = \mathcal{K}_n[\mathrm{Sch}(z, u), \partial_u] = \mathcal{K}_n[a]$. On the other hand, when using a specific configuration for $z(u)$ we can go back to the boundary metric constraint (51) and explicitly solve for $\bar{z}(u)$. Plugging-in this solution together with (60) into the formula for the extrinsic curvature $K[z(u), \bar{z}(u)]$ (53), we can find $\mathcal{K}_n[a]$ and, consequently, find the powers of the Schwarzian in $\mathcal{K}_n[\mathrm{Sch}(z, u), \partial_u]$.

The metric constraint involves solving the first order differential equation

$$-\frac{4a\, e^{au}\bar{z}'}{(e^{au} - \bar{z})^2} = \frac{1}{\varepsilon^2}, \tag{61}$$

whose solution, to all orders in perturbation theory in $\varepsilon$, is given by

$$\bar{z}(u) = e^{au}\left(1 - 2a^2\varepsilon^2 - 2a\varepsilon\sqrt{-1 + a^2\varepsilon^2}\right). \tag{62}$$

We can plug this solution for $\bar{z}(u)$ together with the configuration $z(u) = \exp(au)$ to find that

$$K[z(u) = \exp(au)] = \sqrt{1 - \varepsilon^2 a^2}. \tag{63}$$

Depending on the choice of branch one can reverse the sign of (63) to find that $K[z(u) = \exp(au)] = -\sqrt{1 - \varepsilon^2 a^2}$ which corresponds to the considering the exterior of an $AdS_2$ patch as our surface (instead of a regular $AdS_2$ patch). This is analogous to the contracting branch in of the WDW functional in (38).

---

[13]While (60) is, in fact, a solution to the equation of motion for the Schwarzian theory it is not necessarily a solution to the equation of motion in the theory with finite cutoff.

Consequently, it follows that in a perturbative series in $\varepsilon$ we find:[14]

$$K_{\pm}[z(u)] \;\; = \pm\left(\sqrt{1 + 2\varepsilon^2 \operatorname{Sch}(z,u)} \;+\; \text{derivatives of Sch.}\right), \tag{64}$$

where we find that the quadratic term in $\varepsilon$ for the $+$ branch of (64) agrees with the expansion of $K$ in terms of $\varepsilon$ in JT gravity in asymptotic $AdS_2$ [11] (which found that $K[z(u)] = 1 + \varepsilon^2 \operatorname{Sch}(z,u) + \dots$). The $+$ branch in (64) corresponds to compact patches of $AdS_2$ for which the normal vector points outwards; the $-$ branch corresponds to non-compact surfaces (the complement of the aforementioned $AdS_2$ patches) for which the normal vector is pointing inwards. While the $+$ branch has a convergent path integral for real values of $\phi_r$, for a normal choice of countour for $z(u)$, the path integral of the $-$ branch will be divergent. Even for a potential contour choice for which the path integral were convergent, the $-$ branch is non-perturbatively suppressed by $O(e^{-\int_0^{\beta} du\, \phi_b/\varepsilon}) = O(e^{-1/\varepsilon^2})$. Therefore, for now, we will ignore the effect of this different branch ($-$) and set $K[z(u)] \equiv K_{+}[z(u)]$; we will revisit this problem in section 4 when studying non-perturbative corrections in $\varepsilon$.

In principle, one can also solve for the derivative of the Schwarzian in (64) following a similar strategy to that outlined above. Namely, it is straightforward to find that when $\operatorname{Sch}(z,u) = au^n$, for some $n \in \mathbb{Z}$, then $z(u)$ is related to a Bessel function. Following the steps above, and using the fact that $\partial^{n+1}\operatorname{Sch}(z,u) = 0$ for such configurations, one can then determine all possible terms appearing in the extrinsic curvature. However, since we are interested in quantizing the theory in a constant dilaton configuration, we will shortly see that we can avoid this more laborious process.

Therefore, the JT action that we are interested in quantizing is given by:

$$I_{JT} = -\int_0^{\beta} \frac{du}{\varepsilon^2} \phi_r \left(\sqrt{1 + 2\varepsilon^2 \operatorname{Sch}(z,u)} - 1 + \text{derivatives of Sch.}\right), \tag{65}$$

where we have added the correct counter-term needed in order to cancel the $1/\varepsilon^2$ divergence in the $\varepsilon \to 0$ limit.

While we have found $K[z(u)]$ and $I_{JT}$ to all orders in perturbation theory in $\varepsilon$, we have not yet studied other non-perturbative pieces in $\varepsilon$ (that do not come from the $-$ branch in (64)). Such corrections could contain non-local terms in $u$ since all terms containing a finite number of derivatives in $u$ are captured by the $\varepsilon$-perturbative expansion. The full solution of (61) provides clues that such non-perturbative corrections could exist and are, indeed, non-local (as they will not be a functional of the Schwarzian). The full solution to (61) is

$$\bar{z}(u) = e^{au}\left(1 - 2a^2\varepsilon^2 + 2a\varepsilon\left(\sqrt{-1 + a^2\varepsilon^2} - \frac{2\varepsilon}{\frac{\varepsilon}{\sqrt{-1+a^2\varepsilon^2}} + \mathcal{C}_1 e^{\frac{u}{\varepsilon}\sqrt{-1+a^2\varepsilon^2}}}\right)\right), \tag{66}$$

for some integration constant $\mathcal{C}_1$. When $\mathcal{C}_1 \neq 0$, note that the correction to $\bar{z}(u)$ in (66) are exponentially suppressed in $1/\varepsilon$ and do not contribute to the series expansion $\mathcal{K}_n$. However, when taking $\mathcal{C}_1 \neq 0$, (66) there is no way of making $\bar{z}(u)$ periodic (while it is possible to make $z(u)$ periodic). While we cannot make sure that every solution has the feature that non-perturbative corrections are inconsistent with the thermal boundary conditions, for the remainder of this section we will only focus on the perturbative expansion of $K[z(u)]$ with the branch choice for the square root given by (64). We will make further comments about the nature of non-perturbative corrections in section 4.

---

[14]The terms containing derivatives of the Schwarzian are not necessarily total derivatives and thus we need to explain why they do not contribute to the path integral.

### 3.4 Path integral measure

Before we proceed by solving the path integral of (65), it is important to discuss the integration measure and integration contour for $z(u)$. Initially, before imposing the constraint (51) on the boundary metric, we can integrate over both $z(u)$ and $\bar{z}(u)$, with the two variables being complex conjugates in Euclidean signature. However, once we integrate out $\bar{z}(u)$ we are free to choose an integration contour consistent with the constraint (51) and with the topological requirements discussed at the beginning of this section. Thus, for instance if we choose $z(u) \in \mathbb{R}$ then the constraint (51) would imply that $z'(u) > 0$ (or $z'(u) < 0$); this, in turn, implies that we solely need to integrate over strictly monotonic functions $z(u)$. The boundary conditions for $z(u)$ should nevertheless be independent of the choice of contour; therefore we will impose that $z(u)$ is periodic, $z(0) = z(\beta)$. Of course, this implies that $z(u)$ has a divergence. In order to impose that the boundary is never self-intersecting we will impose that this divergence occurs solely once.[15] Such a choice of contour therefore satisfies the following two criteria:

- That the boundary is not self-intersecting.

- The boundary is time-like when going to Lorentzian signature. This is because redefining $z(u) \to z^{\text{Lor.}}(u) = -iz(u) \in \mathbb{R}$ leaves the action invariant and describes the boundary of a Lorentzian manifold. Since $i(z^{\text{Lor.}})' > 0$, it then follows that the boundary would be time-like.

Furthermore, while we have chosen a specific diffeomorphism gauge which fixes $\gamma_{uu} = 1/\varepsilon^2$, the path integral measure (as opposed to the action) should be unaffected by this choice of gauge and should rather be diffeomorphism invariant. The only possible local diffeomorphism invariant path integral measure is that encountered in the Schwarzian theory [12, 13, 40] and, in JT gravity at infinite cutoff [23]:

$$D\mu[z] = \prod_{z \in [0, \beta)} \frac{dz(u)}{z'(u)}. \tag{67}$$

In principle, one should also be able to derive (67) by considering the symplectic form for JT gravity obtained from an equivalent $\mathfrak{sl}(2, \mathbb{R})$ BF-theory. In [23] this symplectic form (which in turn yields the path integral measure (67)) was derived in the limit $\varepsilon \to 0$. It would however be interesting to rederive the result of [23] at finite $\varepsilon$ in order to find a more concrete derivation of (67).

To summarize, we have therefore argued that both the path integration measure, as well as the integration contour, in the finite-$\varepsilon$ theory, can be taken to be the same as those in the pure Schwarzian theory.

### 3.5 Finite cutoff partition function as a correlator in the Schwarzian theory

The path integral which we have to compute is given by

$$Z_{JT}[\phi_b, L] = \int_{z'(u)>0} D\mu[z] \exp\left[ \int_0^\beta \frac{du}{\varepsilon^2} \phi_r \left( \sqrt{1 + 2\varepsilon^2 \text{Sch}(z, u)} - 1 + \right. \right.$$
$$\left. \left. + \text{ derivatives of Sch.} \right) \right], \tag{68}$$

---

[15]All this is also the case in the Schwarzian theory whose classical solution is $\tau(u) = \tan(\pi u/\beta)$. [11] has found that if considering solutions where $\tau(u)$ diverges multiple times ($\tau(u) = \tan(n\pi u/\beta)$ with $n \in \mathbb{Z}$) then the fluctuations around such solutions are unbounded, and the path integral is divergent (one can still make sense of this theory though, as explained in [24]).

Of course, due to the agreement of integration contour and measure, we can view (68) as the expectation value of the operator in the pure Schwarzian theory with coupling $\phi_r$:

$$Z_{JT}[\phi_b, L] = \langle \mathcal{O}_{\text{deformation}} \rangle \equiv \tag{69}$$

$$\equiv \left\langle \exp\left[ \int_0^\beta \frac{du}{\varepsilon^2} \phi_r \left( \sqrt{1 + 2\varepsilon^2 \text{Sch}(z,u)} - 1 - \varepsilon^2 \, \text{Sch}(z,u) + \text{derivatives of Sch.} \right) \right] \right\rangle.$$

A naive analysis (whose downsides will be mention shortly) would conclude that, since in the pure Schwarzian theory, the Schwarzian can be identified with the Hamiltonian of the theory ($-\frac{H}{2\phi_r^2} = \text{Sch}(z,u)$), then computing (69) amounts to computing the expectation value for some function of the Hamiltonian and of its derivatives. In the naive analysis, one can use that the Hamiltonian is conserved and therefore all derivatives of the Schwarzian in (69) can be neglected. The conservation of the Hamiltonian would also imply that the remaining terms in the integral in the exponent (69) are constant. Therefore, the partition function simplifies to

$$Z_{JT}[\phi_b, L] =_{\text{naive}} \left\langle \exp\left[ \frac{\beta\phi_r}{\varepsilon^2} \left( \sqrt{1 - \frac{\varepsilon^2}{\phi_r^2} H} - 1 + \frac{\varepsilon^2}{2\phi_r^2} H \right) \right] \right\rangle. \tag{70}$$

which can be conveniently rewritten in terms of the actual boundary value of the dilaton $\phi_b = \phi_r/\varepsilon$ and the proper length $L = \beta/\varepsilon$ as

$$Z_{JT}[\phi_b, L] =_{\text{naive}} \left\langle \exp\left[ L\phi_b \left( \sqrt{1 - \frac{H}{\phi_b^2}} - 1 + \frac{H}{\phi_b} \right) \right] \right\rangle. \tag{71}$$

The result for this expectation value in the Schwarzian path integral is given by

$$Z_{JT}[\phi_b, L] =_{\text{naive}} \int ds \, s \sinh(2\pi s) e^{L\phi_b \left( \sqrt{1 - \frac{s^2}{\phi_b^2}} - 1 \right)} \tag{72}$$

where we have identified the energy of the Schwarzian theory in terms of the $\mathfrak{sl}(2,\mathbb{R})$ Casimir for which (for the principal series) $E = C_2(\lambda = is + \frac{1}{2}) + \frac{1}{4} = s^2$ (see [21,22,26,41]). The result (72) agrees with both the result for the WDW wavefunctional presented in section 2 (up to an overall counter-term) and with the results of [9,10] (reviewed in the introduction), obtained by studying an analogue of the $T\bar{T}$ deformation in $1d$.[16]

As previously hinted, the argument presented above is incomplete. Namely, the problem appears because correlation functions of the $\text{Sch}(z,u)$ are not precisely the same as those of a quantum mechanical Hamiltonian. While at separated points correlation functions of the Schwarzian are constant (just like those of $1d$ Hamiltonians), the problem appears at identical points where contact-terms are present. Therefore, the rest of this section will be focused on a technical analysis of the contribution of these contact-terms, and we will show that the final result (72) is indeed correct even when including such terms.

**The generating functional**

To organize the calculation we will first present a generating functional for the Schwarzian operator in the undeformed theory. This generating functional is defined by

$$Z_{\text{Sch}}[j(u)] \equiv \int \frac{D\mu[z]}{SL(2,\mathbb{R})} e^{\int_0^\beta du j(u) \text{Sch}(z(u),u)}, \tag{73}$$

---

[16]We identify the deformation parameter $\lambda = \frac{\varepsilon^2}{4\phi_r}$ in [9,10].

for an arbitrary function $j(u)$ which acts as a source for Schwarzian insertions. This path integral can be computed repeating the procedure in [13], which we also review in appendix A.1. The final answer is given by

$$Z_{\text{Sch}}[j(u)] \sim e^{\int_0^\beta du \frac{j'(u)^2}{2j(u)}} \int ds\, s \sinh(2\pi s) e^{-\frac{s^2}{2} \int_0^\beta \frac{du}{j(u)}} . \tag{74}$$

We will use (74) to evaluate the integrated correlator (69), by rewriting it as

$$\langle \mathcal{O}_{\text{deformation}} \rangle = \left[ \exp\left( \int_0^\beta \frac{du}{\varepsilon^2} \phi_r : \left( \sqrt{1 + 2\varepsilon^2 \frac{\delta}{\delta j(u)}} - 1 + \mathcal{K}\left[ \partial_u \frac{\delta}{\delta j(u)} \right] \right) : \right) \right.$$
$$\left. \times Z_{\text{Sch}}[j(u)] \right]\Bigg|_{j(u)=0} , \tag{75}$$

where $\mathcal{K}\left[ \partial_u \frac{\delta}{\delta j(u)} \right]$ is a placeholder for terms containing derivative terms of the Schwarzian and, equivalently, for terms of the from $\ldots \partial_u \frac{\delta}{\delta j(u)} \ldots$. Finally, $: \mathcal{O} :$ is a point-splitting operation whose role we will clarify shortly.

**Computing the full path integral**

To understand the point splitting procedure necessary in (78), we start by analyzing the structure of correlators when taking functional derivatives of $Z_{JT}[j(u)]$. Schematically, we have that

$$\left( \frac{\delta}{\delta j(u_1)} \cdots \frac{\delta}{\delta j(u_n)} Z_{\text{Sch}}[j(u)] \right)\Bigg|_{j(u)=\phi_r} = a_1 + a_2[\delta(u_{ij})] + a_3[\partial_u \delta(u_{ij})] + \ldots , \tag{76}$$

where $a_1$ is a constant determined by the value of the coupling constant $\phi_r$ and $a_2[\delta(u_{ij})]]$ captures terms which have $\delta$-functions in the distances $u_{ij} = u_i - u_j$, while $a_3[\partial_u \delta(u_{ij})]$ contains terms with at least one derivative of the same $\delta$-functions for each term.[17] The $\ldots$ in (76) capture potential higher-derivative contact-terms.

If in the expansion of the square root in the exponent of (75) one takes the functional derivative $\delta/\delta j(u)$ at identical points then the contact terms in (76) become divergent (containing $\delta(0)$, $\delta'(0)$, $\ldots$). An explicit example about such divergences is given in appendix C when evaluating the contribution of $K_4[z]$ in the perturbative series. In order to eliminate such divergences we define the point-splitting procedure

$$: \frac{\delta^n}{\delta j(u)^n} :\equiv \lim_{(u_1, \ldots, u_n) \to u} \frac{\delta}{\delta j(u_1)} \cdots \frac{\delta}{\delta j(u_n)} . \tag{77}$$

Such a procedure eliminates the terms containing $\delta(0)$ or its derivatives since we first evaluate the functional derivatives in the expansion of (78) at separated points.

The structure of the generating functional also suggests that when integrating the correlator (76) the contribution of the derivatives of $\delta(u_{ij})$ vanish after integration by parts since we will be evaluating (78) for constant dilaton values. As we explain in more detail in appendix C, the origin of the derivatives of $\delta(u_{ij})$ is two-fold: they either come by taking functional derivatives $\delta/\delta j(u)$ of the term $\exp\left( \int_0^\beta du \frac{j'(u)^2}{2j(u)} \right)$ in $Z_{Sch}[j(u)]$, or they come from the contribution of the derivative terms $\mathcal{K}\left[ \partial_u \frac{\delta}{\delta j(u)} \right]$. In either case, both sources only contribute terms containing derivatives of $\delta$-functions (no constant terms or regular $\delta$-functions). Thus, since such

---

[17]For example, when $n = 2$ the exact structure of (76) is computed in [13] and is reviewed in appendix C.

terms vanish after integration by parts, neither $\mathcal{K}\left[\partial_u \frac{\delta}{\delta j(u)}\right]$ nor $\exp\left(\int_0^\beta du \frac{j'(u)^2}{2j(u)}\right)$ contribute to the partition function. Consequently, we have to evaluate

$$
\langle \mathcal{O}_{\text{deformation}} \rangle
$$
$$
= \left( \int ds\, s \sinh(2\pi s) \exp\left[ \int_0^\beta \frac{du}{\varepsilon^2} \phi_r\left( :\sqrt{1+2\varepsilon^2 \frac{\delta}{\delta j(u)}}: -1 \right) \right] e^{-\frac{s^2}{2}\int_0^\beta du \frac{1}{j(u)}} \right)\Bigg|_{j(u)=0}. \quad (78)
$$

To avoid having to deal with the divergences eliminated by the point-splitting discussed in the continuum limit, we proceed by discretizing the thermal circle into $\beta/\delta$ units of length $\delta$ (and will ultimately consider the limit $\delta \to 0$).[18] Divergent terms containing $\delta$ in the final result correspond to terms that contain $\delta(0)$ in the continuum limit and thus should be eliminated by through the point-splitting procedure (77). Therefore, once we obtain the final form of (78), we will select the universal diffeomorphism invariant $\delta$-independent term.

To start, we can use that

$$
e^{-\frac{s^2\delta}{2j(u)}} = \frac{1}{2\pi i} \int_{-c-i\infty}^{-c+i\infty} d\alpha_u \left[ -\frac{\pi Y_1(2\sqrt{\alpha_u})}{\sqrt{\alpha_u}} \right] e^{-\frac{2\alpha_u j_u}{s^2\delta}} \quad (79)
$$

where we have introduced a Lagrange multiplier $\alpha_u$ for each segment in the thermal circle. The integration contours for all $\alpha_u$ are chosen along the imaginary axis for some real constant $c$. The next step is to apply the differential operator in the exponent in (78) to (79),

$$
\left( e^{\int_0^\beta du \frac{\phi_r}{\varepsilon^2}\left( :\sqrt{1+2\varepsilon^2 \frac{\delta}{\delta j(u)}}: -1 \right)} \right) \prod_{u\in[0,\beta)} e^{-\frac{2\alpha_u j_u}{s^2\delta}} \Bigg|_{j_u=0} =
$$
$$
= \left( e^{\int_0^\beta du \frac{\phi_r}{\varepsilon^2}\left( :\sqrt{1+2\varepsilon^2 \frac{\delta}{\delta j(u)}}: -1 \right)} \right) e^{-\int_0^\beta du \frac{2\alpha_u j_u}{s^2\delta^2}} \Bigg|_{j_u=0}
$$
$$
=: \exp\left[ \sum_{u\in[0,\beta)} \frac{\delta\phi_r}{\varepsilon^2}\left( \sqrt{1-\frac{4\alpha_u\varepsilon^2}{s^2\delta^2}} -1 \right) \right] :, \quad (80)
$$

where $: \cdots :$ indicates that we will be extracting the part independent of the UV cutoff, $\delta$, when taking the limit $\delta \to 0$. Thus, we now need to compute

$$
Z_{JT}[\phi_b, L] = \frac{1}{2\pi i} : \int_0^\infty ds\, s \sinh(2\pi s) \quad \int_{-c-i\infty}^{-c+i\infty} \left(\prod d\alpha_u\right) \left[ -\frac{\pi Y_1(2\sqrt{\alpha_u})}{\sqrt{\alpha_u}} \right]
$$
$$
\times e^{\sum_{u\in[0,\beta)} \frac{\delta\phi_r}{\varepsilon^2}\left( \sqrt{1-\frac{4\alpha_u\varepsilon^2}{s^2\delta^2}} -1 \right)} :. \quad (81)
$$

In order to do these integrals we introduce an additional field $\sigma_u$, such that

$$
e^{\frac{\delta\phi_r}{\varepsilon^2}\left( \sqrt{1-\frac{4\alpha_u\varepsilon^2}{s^2\delta^2}} -1 \right)} = \int_0^\infty \frac{d\sigma_u}{\sigma_u^{3/2}} \sqrt{-\frac{\delta\phi_r}{2\pi\varepsilon^2}} e^{-\frac{2\sigma_u\alpha_u\phi_r}{s^2\delta} + \frac{\delta\phi_r}{2\sigma_u\varepsilon^2}(1-\sigma_u)^2}, \quad (82)
$$

where in order for the integral (82) to be convergent, we can analytically continue $\phi_r$ to complex values. We can now perform the integral over $\alpha_u$ using (79), since $\alpha_u$ now appears once again in the numerator of the exponent:

$$
Z_{JT}[\phi_b, L] = : \int_0^\infty ds\, s \sinh(2\pi s)
$$
$$
\times \int_0^\infty \left( \prod_{u\in[0,\beta)} \frac{d\sigma_u}{\sigma_u^{3/2}} \sqrt{-\frac{\delta\phi_r}{2\pi\varepsilon^2}} \right) e^{\sum_{u\in[0,\beta)} \left[ -\frac{s^2\delta}{2\sigma_u\phi_r} + \frac{\delta\phi_r}{2\sigma_u\varepsilon^2}(1-\sigma_u)^2 \right]} :. \quad (83)
$$

---

[18]Sums and products of the type $\sum_{u\in[0,\beta)}$ and $\prod_{u\in[0,\beta)}$ will iterate over all $\beta/\delta$ intervals.

We now change variable in the equation above from $\sigma_u \to 1/\tilde{\sigma}_u$ and perform the Laplace transform, once again using (82). We finally find that (when keeping the finite terms in $\delta$) the partition function is given by:[19]

$$
\begin{aligned}
Z_{JT}[\phi_b, L] \quad &\sim \int_0^\infty ds\, s \sinh(2\pi s)\, e^{\frac{\beta\phi_r}{\varepsilon^2}\left(\sqrt{1-\frac{s^2\varepsilon^2}{\phi_r^2}}-1\right)} \\
&\sim \int_0^\infty ds\, s \sinh(2\pi s)\, e^{\frac{\beta}{4\lambda}\left(\sqrt{1-4\lambda s^2/\phi_r}-1\right)},
\end{aligned}
\tag{84}
$$

where we defined $\lambda = \varepsilon^2/(4\phi_r)$. This partition function agrees with the naive result (72) obtained by replacing the Schwarzian with the Hamiltonian of the pure theory. Consequently, we arrive to the previously mentioned matching between the Euclidean partition function, the WDW wavefunctional and the partition function of the $T\bar{T}$ deformed Schwarzian theory,

$$
e^{-I_{\text{ct}}}\Psi_{HH}[\phi_b, L] = Z_{\lambda=\varepsilon^2/(4\phi_r)}(\beta) = Z_{JT}[\phi_b, L].
\tag{85}
$$

As a final comment, the Euclidean path integral approach hides two ambiguities. First, as we briefly commented in section 3.3, the finite cutoff expansion of the extrinsic curvature might involve terms that are non-perturbatively suppressed in $\varepsilon$. As we have mentioned before, such terms can either come from considering non-local terms in the extrinsic curvature $K[z(u)]$ or by considering the contribution of the negative branch in (64). Second, even if these terms would vanish, the perturbative series is only asymptotic. Performing the integral (84) over energies explicitly gives a finite cutoff partition function

$$
Z_{JT}[\phi_b, L] = \frac{L\phi_b^2 e^{-L\phi_b}}{L^2 + 4\pi^2} K_2\left(-\sqrt{\phi_b^2(L^2 + 4\pi^2)}\right).
\tag{86}
$$

This formal result is not well defined since the Bessel function is evaluated at a branch cut [20]. The ambiguity related to the presence of this branch cut can be regulated by analytic continuation; for example, in $L \to Le^{i\epsilon}$, and the $\epsilon \to 0$ limit we find different answers depending on the sign of $\epsilon$. The ambiguity given by the choice of analytic continuation can be quantified by the discontinuity of the partition function Disc $Z$ for real $\phi$ and $L$.

A similar effect is reproduced by the contracting branch of the wavefunction from the canonical approach, there are two orthogonal solutions to the gravitational constraint $\Psi_{\pm}$, defined by their small cutoff behavior $\Psi_{\pm} \sim e^{\pm\phi L}Z_{\pm}$, where $Z_{\pm}$ is finite. In the language of the Euclidean path integral, the different choice of wavefunctionals correspond to different choices for the square root in the extrinsic curvature (64). Imposing Hartle-Hawking boundary conditions fixes $\Psi_{+}$, which matches the perturbative expansion of the Euclidean path integral. The corrections to the partition function from the other branch are exponentially suppressed $\Psi_{-}/\Psi_{+} \sim e^{-\frac{1}{\varepsilon^2}}$.

As previously hinted, contributions from turning on $\Psi_{-}$ are not only related to the choice of branch for $K[z(u)]$, but is the same as the branch-cut ambiguity mentioned above for (86). To see this, we can notice that Disc $Z$ is a difference of two functions that separately satisfy the WDW equation and goes to zero at small cutoff. Therefore it has to be of the same form as the $\Psi_{-}$ branch given in (38).

---

[19]Once again to integrate over $\tilde{\sigma}_u$ we have to analytically continue $\phi_r$ to complex values. Finally, to perform the integral over $s$ in (84) we analytically continue back to real values of $\phi_r$ and, equivalently, $\phi_b$.

[20]This can be tracked to the fact that we are sitting at a Stokes line. It is curious that this explicit answer gives a complex function even though the perturbative terms we found from the path integral are all real (this phenomenon also happens in more familiar setups like WKB [42]).

# 4 The contracting branch and other topologies

In this section, we will analyze two different kinds of non-perturbative corrections to the partition function. First we will study corrections that are non-perturbative in the cutoff parameter $\varepsilon$ in sections 4.1 and 4.2, which come from turning on the contracting branch of the wavefunction. Then, we will comment on non-perturbative corrections coming from non-trivial topologies in section 4.3.

## 4.1 Unitarity at finite cutoff

Given the exact form of the wavefunction for general cutoff surfaces, we can study some of the more detailed questions about $T\bar{T}$ in $AdS_2$. One such question is whether the theory can be corrected to become unitary. As can be seen from the expression for the dressed energy levels (2), the energies go complex whenever $\lambda > 1/(8E)$. This is unsatisfactory if we want to interpret the finite cutoff JT gravity partition function as being described by a $0+1$ dimensional theory, just like the Schwarzian theory describes the full $AdS_2$ bulk of JT gravity. There are a few ways in which one can go around this complexification.

Firstly, we can truncate the spectrum of the initial theory so that $E$ is smaller than some $E_{\max}$. This is totally acceptable, but if we want to have an initial theory that describes the full $AdS_2$ geometry, we cannot do that without making the flow irreversible. In other words, the truncated Schwarzian partition function is not enough to describe the entire JT bulk. The second option is to accept there are complex energies along the flow but truncate the spectrum to real energies after one has flowed in the bulk. In $1D$ this was emphasized in [9] (and in [1,4] for $2D$ CFTs). The projection operator that achieves such a truncation will then depend on $\lambda$ and, in general, will not solve the flow equation (48) of the partition function. A third option is that we use the other branch of the deformed energy levels $\mathcal{E}_-$ (see (2)) to make the partition function real. In doing so, we will be guaranteed a solution to the Wheeler-de-Witt equation. Let us pursue option three in more detail and show that we can write down a real partition function $Z_\lambda(\beta)$ with the correct (Schwarzian) boundary condition at $\lambda \to 0$.

The solution to the $T\bar{T}$ flow equation (48) that takes the form of a partition function is,

$$Z_\lambda^{\text{non-pert.}}(\beta) = \int_0^\infty dE \rho_+(E) e^{-\beta \mathcal{E}_+(E,\lambda)} + \int_{-\infty}^\infty dE \rho_-(E) e^{-\beta \mathcal{E}_-(E,\lambda)}. \tag{87}$$

Here, we took the ranges of $E$ to be such that $\mathcal{E}_\pm$ are bounded from below. As $\lambda \to 0$, we see that the first term goes to some constant (as we already saw previously), but the second term goes to zero non-perturbatively in $\lambda$ as $e^{-\beta/(2\lambda)}$. From the boundary condition $\lambda \to 0$ we can therefore not fix the general solution, but only $\rho_+(E) = \sinh(2\pi\sqrt{2CE})$. If we demand the partition function to be real, then both integrals over $E$ in (87) should be cutoff at $E = 1/(8\lambda)$ and it will therefore not be a solution to (48) anymore, because the derivatives with respect to $\lambda$ can then act on the integration limit. However, by picking

$$\rho_- = \begin{cases} -\sinh(2\pi\sqrt{2CE}) & 0 < E < \frac{1}{8\lambda} \\ \hat{\rho}(E) & E < 0 \end{cases}, \tag{88}$$

with $\hat{\rho}(E)$ an arbitrary function of $E$, the boundary terms cancel and we obtain a valid solution to (48) and the associated wavefunction $\Psi = e^{L\phi_b} Z$ will solve the WDW equation (33). The final partition function is then given by (see appendix B for details),

$$Z_\lambda^{\text{non-pert.}}(\beta) = \frac{\pi\beta e^{-\frac{\beta}{4\lambda}}}{\sqrt{2\lambda}(\beta^2 + 16C\pi^2\lambda)} I_2\left(\frac{1}{4\lambda}\sqrt{\beta^2 + 16C\pi^2\lambda}\right) + \int_{-\infty}^0 dE \hat{\rho}(E) e^{-\beta \mathcal{E}_-(E,\lambda)}. \tag{89}$$

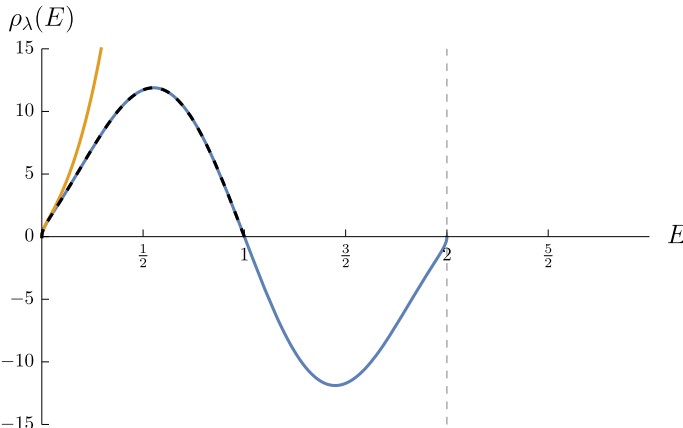

Figure 2: In orange, we show the undeformed density of states $\sinh(2\pi\sqrt{E})$ of JT gravity at infinite cutoff. In dashed black, we show the density of states of the theory with just the branch of the root, $\mathcal{E}_-$, that connects to the undeformed energies, until the energy complexifies. In blue, we show the density of states $\rho_\lambda(E)$ of the deformed partition function (89) which includes non-perturbative corrections in $\lambda$. Above we have set $\hat{\rho}(E) = 0$ and $\lambda = 1/4$ and $C = 1/2$, the black line therefore ends at $E = \frac{1}{4\lambda} = 1$. The vertical dashed line indicates the energy beyond which $\hat{\rho}$ has support.

Notice that when we redefine $E$ such that we have the canonical Boltzman weight in the second term of (89), the support of $\hat{\rho}$ is for $E > \frac{1}{2\lambda}$, because for this redefined energy $E = 0$ maps to $\frac{1}{2\lambda}$. Let us comment on this partition function. First, because of the sign in (88), the first part of (89) has a negative density of states and turns out the be equal to (7) with support between $0 \leq E \leq \frac{1}{2\lambda}$, see Fig. 2. Second, there is a whole function worth of non-perturbative ambiguities coming from the second term in (89) that cannot be fixed by the Schwarzian boundary condition. From the Euclidean path integral approach, assuming that the extrinsic curvature does not receive non-perturbative corrections, we could fix $\hat{\rho}(E) = 0$ by choosing an appropriate analytic continuation on $L$ when defining the partition function.

## 4.2 Relation to $3D$ gravity

The analysis in the previous section can be repeated in the context of $3D$ gravity and $T\bar{T}$ deformations of $2D$ CFTs on a torus of parameters $\tau$ and $\bar{\tau}$. The deformed partition function satisfies an equation similar to (48) derived in [39]. This is given by

$$-\partial_\lambda Z_\lambda = \left[ 8\tau_2 \partial_\tau \partial_{\bar{\tau}} + 4\left( i(\partial_\tau - \partial_{\bar{\tau}}) - \frac{1}{\tau_2} \right) \lambda \partial_\lambda \right] Z_\lambda \tag{90}$$

The solutions of this equation, written in a form of a deformed partition function, can be written as

$$Z(\tau, \bar{\tau}, \lambda) = \sum_{\pm, k} \int_{E_0}^\infty dE \rho_\pm(E) e^{-\tau_2 \mathcal{E}_\pm(E,k) + 2\pi i k \tau_1} \tag{91}$$

where $\tau = \tau_1 + i\tau_2$ and $\bar{\tau} = \tau_1 - i\tau_2$. Here we have set the radius to one and

$$\mathcal{E}_\pm(E,k) = \frac{1}{4\lambda} \left( 1 \mp \sqrt{1 - 8\lambda E + 64\pi^2 k^2 \lambda^2} \right). \tag{92}$$

As usual we pick the minus sign of the root as that connects to the undeformed energy levels at $\lambda = 0$. The energy levels of the deformed partition function complexify when $E_c = \frac{1}{8\lambda} + 8k^2\pi^2\lambda^2$.

So we would like to cutoff the integral there. Similarly, a hard cutoff in the energy will not solve the above differential flow equation anymore. We can resolve this by subtracting the same partition function but with the other sign of the root in (92). This is again a solution, but (again) with negative density of states.

### 4.3 Comments about other topologies

Finally, we discuss the contribution to the path integral of manifolds with different topologies. The contribution of such surfaces is non-perturbatively suppressed by $e^{-\phi_0 \chi(M)}$, where $\chi(M)$ is the Euler characteristic of the manifold.

We start with surfaces with two boundaries of zero genus, where one boundary has the Dirichlet boundary conditions (10) and the other ends on a closed geodesic with proper length $b$. The contribution of such surfaces to the partition function, referred to as "trumpets", has been computed in the infinite cutoff limit in [23]. We can repeat the method of section 2.4 to a spacetime with the geodesic hole of length $b$ by applying the WDW constraints to the boundary on which we have imposed the Dirichlet boundary conditions. This constraint gives the trumpet finite cutoff partition function

$$Z_{\text{trumpet}}[\phi_b, L, b] = \frac{\phi_b L e^{-\phi_b L}}{\sqrt{L^2 - b^2}} K_1\left(-\sqrt{\phi_b^2(L^2 - b^2)}\right). \tag{93}$$

The partition function diverges as $L \to b$, indicating the fact that the boundary with Dirichlet boundary conditions overlaps with the geodesic boundary.

In order to construct higher genus surfaces or surfaces with more Dirichlet boundaries one can naively glue the trumpet to either a higher genus Riemann bordered surface or to another trumpet. In order to recover the contribution to the partition function of such configurations we have to integrate over the closed geodesic length $b$ using the Weil-Petersson measure, $d\mu[b] = db\, b$. However, if integrating over $b$ in the range from 0 to $\infty$ for a fixed value of $L$ we encounter the divergence at $L = b$.

One way to resolve the appearance of this divergence is to once again consider the non-perturbative corrections in $\varepsilon$ discussed in section 4.1 for the trumpet partition function (93). We can repeat the same procedure as in 4.1 by accounting for the other WDW branch thus making the density of states of the "trumpet" real. Accounting for the other branch we find that

$$Z_{\text{trumpet}}^{\text{non-pert.}}[\phi_b, L, b] = \frac{2\pi \phi_b L e^{-\phi_b L}}{\sqrt{L^2 - b^2}} I_1\left(\sqrt{\phi_b^2(L^2 - b^2)}\right), \tag{94}$$

where we set the density of states for negative energies for the contracting branch to 0. Interestingly, the partition function (94) no longer has a divergence at $L = b$ which was present in (93) and precluded us previously from performing the integral over $b$. We could now integrate [21]

$$Z_{\text{cyl.}}^{\text{non-pert.}}[\phi_{b_1}, L_1, \phi_{b_2}, L_2] =_{\text{naive}} \int_0^\infty db\, b\, Z_{\text{trumpet}}^{\text{non-pert.}}[\phi_{b_1}, L_1, b] Z_{\text{trumpet}}^{\text{non-pert.}}[\phi_{b_2}, L_2, b], \tag{95}$$

to obtain a potential partition function for the cylinder.[22]

---

[21]Alternatively, one might hope to directly use WDW together with the results of [23] for arbitrary genus to directly compute the partition function at finite cutoff. However, as pointed out in [43], the WDW framework is insufficient for such a computation; instead, computing the full partition function requires a third-quantized framework which greatly complicates the computation.

[22] While unfortunately we cannot compute the integral over $b$ exactly it would be interesting to check whether the partition function for the cylinder can be reproduced by a matrix integral whose leading density of states is given by the one found from the disk contribution.

Besides the ambiguity related to the non-perturbative corrections, there is another issue with the formula for the cylinder partition function (95). Specifically, for any value of the proper length $L_1$ and $L_2$ and for a closed geodesic length $b$ (with $b < L_1$ and $b < L_2$) there exist cylinders for which the Dirichlet boundaries intersect with the closed geodesic of length $b$. Such surfaces cannot be obtained by gluing two trumpets along a closed geodesic as (95) suggests when using the result (93). Given that the partition function (94) does not have a clear geometric interpretation when including the contributions from the contracting branch, it is unclear if (95) accounts for such geometries. Given these difficulties, we hope to revisit the problem of summing over arbitrary topologies in the near future.

As another example of non-trivial topology, one can study the finite cutoff path integral in a disk with a conical defect in the center. Such defects were previously studied at infinite cutoff in [24]. The answer from the canonical approach is given by

$$Z_{\text{defect}}[\phi_b, L] = \frac{\phi L}{\sqrt{L^2 + 4\pi^2 \alpha^2}} K_1\left(-\sqrt{\phi_b^2(L^2 + 4\pi^2 \alpha^2)}\right), \tag{96}$$

where $\alpha$ is the opening angle, and $\alpha = 1$ gives back the smooth disk wavefunction. This function is finite for all $L$.

## 5 de Sitter: Hartle-Hawking wavefunction

As a final application of the results in this paper, we will study JT gravity with positive cosmological constant, in two-dimensional nearly $dS$ spaces. We will focus on the computation of the Hartle-Hawking wavefunction, see [33], and [44]. The results in these references focus on wavefunctions at late times, with an accurate Schwarzian description. Using the methods in this paper, we will be able to compute the exact wavefunction at arbitrary times.

The Lorentzian action for positive cosmological constant JT gravity is given by

$$I_{\text{JT}} = \frac{1}{2} \int_M \sqrt{g} \phi(R - 2) - \int_{\partial M} \sqrt{\gamma} \phi K. \tag{97}$$

Following section 2.3, we use the ADM decomposition of the metric

$$ds^2 = -N^2 dt^2 + h(d\theta + N_\perp dt)^2, \quad h = e^{2\sigma} \tag{98}$$

where now $t$ is Lorentzian time and $\theta$ the spatial direction. We will compute the wavefunction of the universe $\Psi[L, \phi_b(u)]$ as a function of the total proper length of the universe $L$ and the dilaton profile $\phi_b(u)$ along a spatial slice. The proper spatial length along the boundary is defined by $du = e^\sigma d\theta$. The solution satisfying the gravitational constraints is given by

$$\Psi_+[\phi_b(u), L] = \int dM \rho(M) e^{-i \int_0^L du \left[\sqrt{\phi_b^2 - M + (\partial_u \phi_b)^2} - \partial_u \phi_b \tanh^{-1}\left(\sqrt{1 + \frac{\phi_b^2 - M}{(\partial_u \phi_b)^2}}\right)\right]}, \tag{99}$$

where the index $+$ indicates we will focus on the expanding branch of the wavefunction. This is defined by its behavior $\Psi_+ \sim e^{-i \int_0^L du\, \phi_b(u)}$ in the limit of large universe (large $L$).

To get the wavefunction of the universe, we need to impose the Hartle-Hawking boundary condition. We will look again to the limit of large $L$, and for simplicity, we can evaluate it for a constant dilaton setting $\partial_u \phi_b = 0$ (this is enough to fix the expanding branch of the wavefunction completely).

As explained in [33], one can independently compute the path integral with Hartle-Hawking boundary conditions in this limit by integrating out the dilaton first. This fixes the geometry to

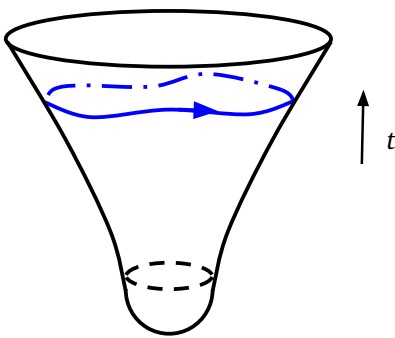

Figure 3: Frame in which geometry is rigid $dS_2$. Time runs upwards. We show the wiggly curve where we compute the wavefunction in blue (defined by its length and dilaton profile).

be rigid $dS_2$, up to the choice of embedding of the boundary curve inside rigid $dS_2$, see figure 3. Then the result reduces to a Schwarzian path integral parametrizing boundary curves, just like in $AdS_2$. The final result for a constant dilaton and total length $L$ is given by

$$\Psi_+[\phi_b, L] \sim e^{-i\phi_b L} \int dM \sinh(2\pi\sqrt{M}) e^{iL\frac{M}{2\phi_b}}, \quad L\phi_b \to \infty, \quad \phi_b/L \text{ fixed.} \tag{100}$$

This boundary condition fixes the function $\rho(M)$ in (99), analogously to the procedure in section 2.4.

Then the final answer for the expanding branch of the Hartle-Hawking wavefunction of JT gravity is

$$\Psi_+[\phi_b, L] = \int_0^\infty dM \sinh(2\pi\sqrt{M}) e^{-i\int_0^L du\left[\sqrt{\phi_b^2 - M + (\partial_u\phi_b)^2} - \partial_u\phi_b \tanh^{-1}\left(\sqrt{1 + \frac{\phi_b^2 - M}{(\partial_u\phi_b)^2}}\right)\right]}. \tag{101}$$

The same result can be reproduced for constant values of $\phi_b(u) = \phi_b$ by following the procedure in section 3, writing the extrinsic curvature along the spatial slice as a functional of the Schwarzian derivative. Following the same steps as in section 3.5, one could then recover the wavefunction (101) by computing the Lorentzian path integral exactly, to all orders in cutoff parameter $\varepsilon$.

The procedure outlined so far parallels the original method of Hartle and Hawking [37]. First, we solve the WDW equation, which for this simple theory can be done exactly. Then, we impose the constraints from the no-boundary condition. The only subtlety is that, while Hartle and Hawking impose their boundary conditions in the past, we are forced to impose the boundary condition at late times. This is a technical issue since the limit $L \to 0$ is strongly coupled. Nevertheless, we could, in principle, do it at early times if we would know the correct boundary condition in that regime.

A different procedure was proposed by Maldacena [45]. The idea is to compute the no-boundary wavefunction by analytic continuation, where one fills the geometry with '$-AdS$' instead of $dS$.[23] We can check now in this simple model that both prescriptions give the same result. For simplicity, after fixing the dilaton profile to be constant, one can easily check that the result (101) found following Hartle and Hawking matches with the analytic continuation of the finite cutoff Euclidean path integral in AdS computed in section 3.

For a constant dilaton profile, we can perform the integral to compute the wavefunction

$$\Psi_+[\phi_b, L] = \frac{L\phi_b^2}{L^2 - 4\pi^2 - i\epsilon} K_2\left(i\sqrt{\phi_b^2(L^2 - 4\pi^2 - i\epsilon)}\right), \tag{102}$$

---

[23]For a review in the context of JT gravity see section 2.3 of [33].

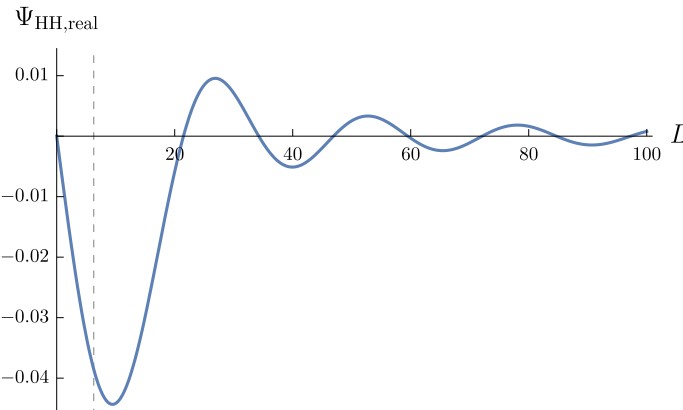

Figure 4: Plot of the wavefuntion $\Psi_{\mathrm{HH,real}}$ for $\phi_b = 1/4$. The vertical dashed line indicates the location, $L = 2\pi$, where the expanding branch $\Psi_+$ of the wavefunction (102) diverges, but $\Psi_{\mathrm{HH,real}}$ remains finite.

where the $i\epsilon$ prescription is needed to make the final answer well defined (see also section 4). This wavefunction satisfies the reduced WDW equation [24]

$$(L\phi - L\partial_L(L^{-1}\partial_\phi))\Psi[L, \phi] = 0. \tag{103}$$

One interesting feature of this formula is the fact that it also satisfies the naive no-boundary condition since $\Psi_+[L \to 0, \phi_b] \to 0$. Nevertheless, even though it behaves as expected for small lengths, it has a divergence at $L_{\mathrm{div}} = 2\pi$ (the Bessel function blows up near the origin). Semiclassically, the geometry that dominates the path integral when $L = 2\pi$ is the lower hemisphere of the Euclidean $S^2$ (dashed line in figure 3). This is reasonable from the perspective of the JT gravity path integral since this boundary is also a geodesic, but it would be nice to understand whether this divergence is unique to JT gravity, or would it also be present in theories of gravity in higher dimensions.

We can also comment on the $T\bar{T}$ interpretation of dS gravity. For large $L$ it was argued in [33] that a possible observable in a dual QM theory computing the wavefunction can be $\Psi_+[L] \sim \mathrm{Tr}[e^{iLH}]$, with an example provided after summing over non-trivial topologies by a matrix integral (giving a dS version of the AdS story in [23]). We can extend this (before summing over topologies) to a calculation of the wavefunction at finite $L$ by $T\bar{T}$ deforming the same QM system. This is basically an analytic continuation of the discussion for $AdS_2$ given in previous sections.

So far we focused on the expanding branch of the wavefunction following [33]. We can also find a real wavefunction analogous to the one originally computed by Hartle and Hawking [37], which we will call $\Psi_{\mathrm{HH,real}}$. This is easy to do in the context of JT gravity and the answer is

$$\Psi_{\mathrm{HH,real}}[L, \phi_b] = \frac{\pi L \phi_b^2}{L^2 - 4\pi^2} I_2\left(i\sqrt{\phi_b^2(L^2 - 4\pi^2)}\right) \tag{104}$$

This wavefunction is real, smooth at $L = 2\pi$ and also satisfies $\Psi_{\mathrm{HH,real}}[L \to 0, \phi_b] \to 0$. We plotted the wavefunction in Fig. 4. For large universes this state has an expanding and contracting branch with equal weight.

Finally, the results of this section can be extended to pure $3D$ gravity with positive cosmological constant $\Lambda = 2/\ell^2$. Using Freidel reconstruction kernel, the wavefunction $\Psi[e^\pm]$

---

[24]This differs from the wavefunction written in [33] since we found a modification in the WDW equation. The solutions are related by $\Psi_{\mathrm{here}} = L\Psi_{\mathrm{there}}$. The Klein-Gordon inner product defined in [33] should also be modified accordingly.

satisfying WDW, as a function of the boundary frame fields $e^{\pm}$, is given by

$$\Psi_+[e^{\pm}] = e^{i\frac{\ell}{16\pi G_N}\int e} \int \mathcal{D}E \; e^{-i\frac{\ell}{8\pi G_N}\int E^+ \wedge E^-} Z(E+e). \tag{105}$$

This is the most general, purely expanding, solution of WDW up to an arbitrary function of the boundary metric $Z(E)$. We can fix $\Psi_+$ uniquely by looking at the late time limit, or more accurately, boundary metrics with large volume. In this limit Freidel formula gives $\Psi_+[Te^{\pm}] \sim e^{iS_{\text{c.t.}}(T,e)}Z(e)$ for large $T$. The first term is rapidly oscillating with the volume $T$ at late times and we see the finite piece is precisely the boundary condition we need $Z(e)$. The path integral calculation of the finite piece $Z(e)$ was done in [44] for the case of a boundary torus (see their equation 4.121 and also [46]) and gives a sum over $SL(2,\mathbb{Z})$ images of a Virasoro vacuum character. We leave the study of the properties of this wavefunction for future work.

# 6 Discussion

JT gravity serves as an essential toolbox to probe some universal features of quantum gravity. In the context of this paper, we have shown that the WDW wavefunctional at finite cutoff and dilaton value in $AdS_2$ agrees with an explicit computation of the Euclidean path integral; this, in turn, matches the partition function of the Schwarzian theory deformed by a $1D$ analog of the $T\bar{T}$ deformation. Consequently, our computation serves as a check for the conjectured holographic duality between a theory deformed by $T\bar{T}$ and gravity, in $AdS$, at a finite radial distance.

*Finite cutoff unitarity*

Beyond providing a check, our computations indicate paths to resolve several open problems related to this conjectured duality. One such issue is that of complex energies that were present when deforming by $T\bar{T}$ (both in 1 and 2$D$), and were also present in the WDW wavefunctional when solely accounting for the expanding branch. However, from the WDW perspective, one could also consider the contribution of the contracting branch, and, equivalently, in the Euclidean path integral, one could also account for the contribution of non-compact geometries. In both cases, such corrections are non-perturbative in the cutoff parameter $\varepsilon$ or, in the context of $T\bar{T}$, in the coupling of the deformation $\lambda$. Nevertheless, we have shown that there exists a linear combination between the two wavefunctional branches that leads to a density of states which is real for all energies. Thus, this suggests that a natural resolution to the problem of complex energy levels is the addition of the other branch, instead of the proposed artificial cutoff for the spectrum once the energies complexify [4]. While the problem of complex energy levels is resolved with the addition of the contracting branch, a new issue appears: the partition function now has a negative density of states. This new density of states implies that, even with such a resolution, the partition function is not that of a single unitary quantum system. In three bulk dimensions, one has a similar state of affairs. The energy levels again complexify, and the other branch of the solution space can cure this, with the caveat that the density of states will become negative. A possible resolution consistent with unitarity would be that the finite cutoff path integral is not computing a boundary partition function but something like an index, where certain states are weighted with a negative sign.

A related issue that leads to the ambiguity in the choice of branches is that the non-perturbative piece of the partition function that cannot be fixed by the $\lambda \to 0$ boundary condition. This ambiguity can be cured by putting additional conditions on the partition function.

Fixing the $\lambda$-derivative of $Z_\lambda^{\text{non-pert.}}(\beta)$ does not work, but for instance $Z_\lambda^{\text{non-pert.}}(\beta) \to 0$ as $\beta \to 0$ would be enough to fix the partition function completely. One other possibility, motivated by the bulk, is to fix the extrinsic curvature $K$ at $\varepsilon \to 0$. This will eliminate one of the two branches and, therefore, also $\hat{\rho}$ in (89).[25] One can also try to foliate the spacetime with different slices, for instance, by taking constant extrinsic curvature slices.[26] In 3$D$, this was done explicitly in [48] for a toroidal boundary and in [49] for more general Riemann surfaces. In particular, for the toroidal boundary, it was found that the wavefunction in the mini-superspace approximation inherits a particular modular invariance, and it would be interesting to compare that analysis to the one done in [39].

In the AdS$_3$/CFT$_2$ context, it would also be interesting to understand the non-perturbative corrections to the partition function purely from the field theory. As the $T\bar{T}$ deformation is a particular irrelevant coupling, it is not unreasonable to suspect that such corrections are due to instanton effects contributing at $O(e^{-1/\lambda})$. The fate of such instantons can be studied using, for example, the kernel methods [32, 50] or the various string interpretation of $T\bar{T}$ [51, 52]; through such an analysis, one could hope to shed some light on the complexification of the energy levels.

### *Application: Wavefunction of the universe*

The techniques presented in this paper also apply to geometries with constant positive curvature. We do this calculation in two ways. On one hand we solve the WDW constraint that this wavefunction satisfies, imposing the Hartle-Hawking boundary condition. On the other hand, we compute the wavefunction as an analytic continuation from the Euclidean path integral on '-AdS'. As expected, we find that both results match. We also analyze two possible choices to define the wavefunction. The first solely includes the contribution of the expanding branch and has a pole when the size of the universe coincides with the dS radius. The second is a real wavefunctional, which includes the non-perturbative contribution of the contracting branch and is now smooth at the gluing location. It would be interesting to identify whether this divergence is present in higher dimensions or if it is special to JT gravity. We also leave for future work a better understanding of the appropriate definition of an inner product between these states.[27] Finally, we outlined how a similar analysis can be used to find the no-boundary wavefunction for pure 3$D$ gravity with a positive cosmological constant, the simplest example corresponding to a toroidal universe.

### *Sum over topologies*

An important open question that remains unanswered is the computation of the JT gravity partition function when including the contribution of manifolds with arbitrary topology. While we have determined the partition function of finite cutoff trumpets using the WDW constraint, this type of surface is insufficient for performing the gluing necessary to obtain any higher genus manifold with a fixed proper boundary length. It would be interesting to understand whether the contribution of such manifolds to the path integral can be accounted for by using an alternative gluing procedure that would work for any higher genus manifold.

For the cylinder, we can actually avoid the gluing. From a third quantisation point of view, one way to think about the cylinder partition function, or double trumpet, is as the propagator associated to the WDW equation in mini-superspace,

$$\left[-L\phi + L\partial_L(L^{-1}\partial_\phi)\right]\Psi_{\text{cylinder}}(\phi, \phi', L, L') = \delta(L - L')\delta(\phi - \phi'). \tag{106}$$

---

[25]However, such a resolution appears to bring back the complex energies.

[26]Appendix A.2, in fact, provides a non-trivial check of the form of the extrinsic curvature $K[z(u)]$ by considering boundary conditions with fixed extrinsic curvature slices. We will provide further comments about such boundary conditions in [47].

[27] In the limit of large universes, some progress in this direction was made in [33].

This avoids the integral over $b$ and since the WDW equation (106) is just the propagator of a massive particle in a constant electric field[28], we can solve it with standard methods. The resulting propagator is proportional to a Hankel function of the geodesic distance on mini-superspace, but does not have the same form as the double trumpet computed in [23] once $L, L', \phi$ and $\phi'$ are taken large. In fact, it vanishes in that limit. Furthermore, there is a logarithmic divergence when the geodesic distance in mini-superspace vanishes, i.e. when $L = L'$ and/or $\phi = \phi'$. There are several reasons for this discrepancy. The obvious one would be that the cylinder is not the propagator in third quantisation language, but this then raises the question, what is this propagator? Does it have a geometric interpretation? It would be interesting to understand this discrepancy better and what the role of the third quantised picture is.

*Coupling to matter* & *generalizations*

Finally, it would be interesting to understand the coupling of the bulk theory to matter. When adding gauge degrees of freedom to a $3D$ bulk and imposing mixed boundary conditions between the graviton and the gauge field, the theory is dual to a $2D$ CFT deformed by the $J\bar{T}$ deformation [53].[29] In $2D$, the partition function of the theory coupled to gauge degrees of freedom can be computed exactly even at finite cutoff; this can be done by combining the techniques presented in this paper with those in [54] [30] . It would be interesting to explore the possibility of a $1D$ deformation, analogous to the $J\bar{T}$ deformation in $2D$, which would lead to the correct boundary dual for the gravitational gauge theory. Since gauge fields do not have any propagating degrees of freedom in $2D$, it would also be interesting to explore the coupling of JT gravity to other forms of matter.[31] In the usual finite cutoff $AdS_3/T\bar{T}$ deformed CFT correspondence, adding matter results in the dual gravitational theory having mixed boundary conditions for the non-dynamical graviton [57]. Only when matter fields are turned off are these mixed boundary conditions equivalent to the typical finite radius Dirichlet boundary conditions. In $2D$ this was done for the matterless case in [10] and it would be interesting to generalise this to include matter.

# Acknowledgements

We thank Alexandre Belin, Steve Giddings, Henry Lin, Juan Maldacena, Don Marolf, Mark Mezei, Onkar Parrikar, Eric Perlmutter, Silviu Pufu, Ronak Soni, Eva Silverstein, Douglas Stanford, Edward Witten and Zhenbin Yang for valuable discussions. Special thanks go to Edgar Shaghoulian for comments on a draft. JK is supported by the Simons Foundation. LVI is supported in part by the US NSF under Grant No. PHY-1820651 and by the Simons Foundation Grant No. 488653. GJT is supported by a Fundamental Physics Fellowship. The research of HV is supported by NSF grant PHY-1620059. This research was supported in part by Perimeter Institute for Theoretical Physics. Research at Perimeter Institute is supported by the Govern-

---

[28]In the coordinates $u = \phi^2$ and $v = L^2$, (33) reduces to $\left(\partial_u \partial_v + \frac{1}{4} - \frac{1}{2v}\partial_u\right)\Psi = 0$. This is the KG equation for $m^2 = 1$ and external gauge field $A = \frac{i}{v}dv$. Notice that the mini-superspace is Lorentzian, whereas the geometries $\Psi$ describes are Euclidean.

[29]Here, $J\bar{T}$ is a composite operator containing $J$, a chiral $U(1)$ current, and $\bar{T}$, a component of the stress tensor.

[30]Another possible direction could be to understand the result for $2D$ gravity as a limit of $3D$ (either for near extremal states [31] or in relation to SYK-like models [55]).

[31]One intriguing possibility is to couple JT gravity to a $2D$ CFT. The effect of the CFT on the partition function has been studied in [22, 56] through the contribution of the Weyl anomaly in the infinite cutoff limit. It would be interesting to see whether the effect of the Weyl anomaly can be determined at finite cutoff solely in terms of the light-cone coordinate $z(u)$.

ment of Canada through the Department of Innovation, Science and Economic Development and by the Province of Ontario through through the ministry of Research and Innovation.

# A   Additional checks

## A.1   WDW with varying dilaton

In this section we will check our formula (45) in the case of a varying dilaton with an arbitrary profile $\phi_b(u)$. We will still work in the limit of large $L$ and $\phi_b$ such that we are working near the boundary of $AdS_2$. Expanding the solution of the WDW equation gives

$$\Psi_{\mathrm{HH}}[\phi_b(u), L] = \int dM \rho_{\mathrm{HH}}(M)\, \exp\left[\int_0^L du\left(\phi_b - \frac{M}{2\phi_b} + \frac{(\partial_u\phi_b)^2}{2\phi_b} + \dots\right)\right] \qquad (107)$$

where the dots denote terms that are subleading in this limit. The first term produces the usual divergence piece $\int_0^L du\,\phi_b(u)$. The second term after integrating over $M$ would produce the Schwarzian partition function with an effective length given by $\ell = \int_0^L \frac{du}{\phi_b(u)}$, which can be interpreted as a renormalized length. The final answer is then

$$\Psi_{\mathrm{HH}}[L, \phi] = e^{\int_0^L du\,\phi_b(u)} Z_{\mathrm{Sch}}\left(\int_0^L \frac{du}{\phi_b(u)}\right) e^{\frac{1}{2}\int_0^L du\frac{(\partial_u\phi_b)^2}{\phi_b}}. \qquad (108)$$

Now we will show the full answer, including the last term in (108), $\frac{1}{2}\int_0^L du\frac{(\partial_u\phi)^2}{\phi}$, can be reproduced by the Euclidean path integral through the Schwarzian action.

For a varying dilaton the bulk path integral of JT gravity can be reduced to

$$\int \mathcal{D}g\mathcal{D}\phi\; e^{-I_{\mathrm{JT}}[\phi, g]} \to e^{\int_0^L du\phi_b(u)}\int \frac{\mathcal{D}f}{SL(2, \mathbb{R})}\, e^{\int_0^L du\phi_b(u)\,\mathrm{Sch}(F(u), u)}, \quad F = \tan\pi f \qquad (109)$$

For simplicity we will assume that $\phi_b(u) > 0$. Following [13] we can compute this path integral using the composition rule of the Schwarzian derivative

$$\mathrm{Sch}(F(\tilde{u}(u)), u) = \mathrm{Sch}(F, \tilde{u})(\partial_u\tilde{u})^2 + \mathrm{Sch}(\tilde{u}, u). \qquad (110)$$

We can pick the reparametrization to be $\partial_u\tilde{u} = 1/\phi_b(u)$. This implies in terms of the coordinate $\tilde{u}$ the total proper length is given by $\tilde{L} = \int_0^L du/\phi_b(u)$. This simplifies the Schwarzian term and we can write the second term as

$$\int_0^L du\,\phi_b(u)\,\mathrm{Sch}(\tilde{u}, u) = \frac{1}{2}\int_0^L du\frac{(\partial_u\phi_b)^2}{\phi_b} \qquad (111)$$

up to total derivative terms that cancel thanks to the periodicity condition of the dilaton. Then we can rewrite the path integral as

$$\int \mathcal{D}g\mathcal{D}\phi\; e^{-I_{\mathrm{JT}}[\phi, g]} \;\;\to\;\; e^{\int_0^L du\,\phi_b(u) + \frac{1}{2}\int_0^L du\frac{(\partial_u\phi_b)^2}{\phi_b}}\int \frac{\mathcal{D}f}{SL(2, \mathbb{R})}\, e^{\int_0^{\tilde{L}} d\tilde{u}\,\mathrm{Sch}(F, \tilde{u})}, \qquad (112)$$

$$= e^{\int_0^L du\,\phi_b(u) + \frac{1}{2}\int_0^L du\frac{(\partial_u\phi_b)^2}{\phi_b}} Z_{\mathrm{Sch}}\left(\tilde{L} = \int_0^L \frac{du}{\phi_b}\right) \qquad (113)$$

which matches with the result coming from the WDW wavefunction (108). This is a nontrivial check of our proposal that $\Psi_{\mathrm{HH}}$ in (45) computes the JT gravity path integral at finite cutoff.

## A.2  JT gravity with Neumann boundary conditions

To provide a further check of the form of the extrinsic curvature $K$ at finite cutoff (65), we can study the theory with Neumann boundary conditions, when fixing the extrinsic curvature $K[z(u)] = K_b$ instead of the boundary dilaton value $\phi_r$ and when fixing the proper length $L$ to be finite in both cases.[32] We will work in Poincaré coordinates (50). Since $K_b > 0$ it means (in our conventions) that we are considering a vector encircling a surface with genus 0 (normal vector pointing outwards). On the Poincaré plane, curves of constant $K_b$ are circles, semi-circles (that intersect the $H_2$ boundary) or lines. All of them can be parametrized in the Poincaré boundary coordinates $\tau(u)$ and $x(u)$ as:

$$\tau(u) = a + b\cos(u), \qquad x(u) = d + b\sin(u), \qquad K_b = \frac{d}{b}, \qquad \sqrt{\gamma_{uu}} = \frac{b}{d + b\sin u}, \quad (114)$$

with $b, d \in \mathbb{R}$. Note that if we want the circle above to be fully contained within the Poincaré half-plane (with $x > 0$) we need to require that $d > 0$ and $d \geq b$ which implies $K_b \geq 1$. Thus, for contractible boundaries which contain the surface inside of them we must have $K_b \geq 1$.

For this value of $K_b$, the boundary proper length is restricted to be

$$\frac{\beta}{\varepsilon} = \int du \sqrt{\gamma_{uu}} = \frac{2\pi}{\sqrt{(K_b + 1)(K_b - 1)}}. \quad (115)$$

Therefore, the partition function with Neuman boundary conditions should solely isolate configurations which obey (115). A non-trivial check will be to recover this geometric constraint by going from the partition function with Dirichlet boundary conditions (for which we obtained the action (65)) and the partition function with Neumann boundary conditions.

In the phase space of JT gravity $K[z(u)]$ and $\phi(u)$ are canonical conjugate variables on the boundary. Therefore, in order to switch between the two boundary conditions at the level of the path integral, we should be able to integrate out $\phi_r(u)$ to obtain the partition function with Neumann boundary conditions. Explicitly we have that,[33]

$$Z_N[K_b(u), L] = \int_{\tilde{\phi}_b - i\infty}^{\tilde{\phi}_b + i\infty} D\phi_b(u) Z_{JT}[\phi_b(u), L] e^{\frac{1}{\varepsilon} \int_0^\beta du\, \phi_b(u)(1 - K_b(u))}$$

$$= \int_{\tilde{\phi}_b - i\infty}^{\tilde{\phi}_b + i\infty} D\phi_b(u) \int D\phi Dg_{\mu\nu} e^{\phi_0 \chi(\mathcal{M}) - S_{\text{bulk}}[\phi, g_{\mu\nu}] + \frac{1}{\varepsilon} \int du\, \phi_b(u)(K - K_b(u))}$$

$$\sim \int D\phi Dg_{\mu\nu} e^{\phi_0 \chi(\mathcal{M}) - S_{\text{bulk}}[\phi, g_{\mu\nu}]} \prod_{u \in \partial \mathcal{M}} \delta(K(u) - K_b(u)). \quad (116)$$

which of course fixes the extrinsic curvature on the boundary. To simplify our computation, we will work with the "renormalized" extrinsic curvature $K_{b,r}$, defined as $K_b \equiv 1 + \varepsilon^2 K_{b,r}$ and choose a constant value for $K_{b,r}$.

Using the formula (65) for $K[z(u)]$ in (116) we can rewrite the second line in terms of a path integral for the Schwarzian mode $z(u)$:

$$Z_N\left[K_b = 1 + \varepsilon^2 K_{b,r}, L = \beta/\varepsilon\right] = \int \frac{d\mu[z(u)]}{SL(2,\mathbb{R})} \prod_{u \in \partial \mathcal{M}} \delta\left(\sqrt{1 + 2\varepsilon^2 \text{Sch}(z(u), u)} - 1 - \varepsilon^2 K_{b,r}\right.$$

$$\left. + \text{derivatives of Sch.}\right). \quad (117)$$

One set of solutions for which the $\delta$-function in (117) are the configurations for which the Schwarzian is a constant (related to $K_{b,r}$) for which all the derivatives of the Schwarzian vanish.[34] Specifically, for such configurations which obey $z(0) = z(\beta)$, we have that $z(u) = \tan(\pi u/\beta)$,

---

[32]A more detailed analysis of the theory with such boundary conditions will be presented in [47].

[33]Where $\tilde{\phi}_b$ is some arbitrary constant which is used to shift the contour along the real axis.

[34]It is possible that there are other solutions which we do not account for in (117) that do not have $\text{Sch}(z(u), u)$ constant but have the sum between the non-derivative terms and derivative terms in (117) still yield the overall constant $1 + \varepsilon^2 K_{b,r}$. While we do not analyze the possible existence of these configuration, it is intriguing that they do not affect the result of (118). We will once again ignore non-perturbative corrections in $\varepsilon$.

which yields:

$$\sqrt{1 + 4\varepsilon^2 \frac{\pi^2}{\beta^2}} - 1 = \varepsilon^2 K_{b,r} \qquad \Rightarrow \qquad \frac{\beta}{\varepsilon} = \frac{2\pi}{\sqrt{\varepsilon^2 K_{b,r}(2 + \varepsilon^2 K_{b,r})}} = \frac{2\pi}{\sqrt{(K_b + 1)(K_b - 1)}} \tag{118}$$

which exactly matches the constraint (115). This is a strong consistency check that the relation between the deformed Schwarzian action (65) and the extrinsic curvature when moving to finite cutoff.

## B  General solution to (48)

In this appendix we present a more general analysis of the differential equation (48), which we reproduce here for convenience,

$$\left[4\lambda\partial_\lambda\partial_\beta + 2\beta\partial_\beta^2 - \left(\frac{4\lambda}{\beta} - 1\right)\partial_\lambda\right] Z_\lambda(\beta) = 0. \tag{119}$$

In particular, since (4) appears (at least naively) to not converge and the integral transform (5) is not well-defined for the sign of $\lambda$, i.e. $\lambda > 0$, which is appropriate for JT gravity at finite cutoff, the solution to the differential equation provides a solution for the partition function for that sign.

To solve the differential equation (119) it is useful to decouple $\lambda$ and $\beta$. This can be done by defining $R = \beta/(8\lambda)$ and $e^\sigma = \beta/(2C)$ and writing the problem in terms of $R$ and $\sigma$. The differential equation becomes,

$$-R^2(\partial_R^2 + 4\partial_R)Z + (\partial_\sigma^2 - \partial_\sigma)Z = 0 \tag{120}$$

By using separation of variables we find that the general solution is,

$$Z(R,\sigma) = \int_{-\infty}^{\infty} d\nu\, e^{-\nu\sigma}\sqrt{R}\, e^{-2R}\left(a_\nu K_{1/2+\nu}(-2R) + b_\nu K_{1/2+\nu}(2R)\right) \tag{121}$$

where $\nu$ is the related to the seperating contant. We are interested in find the solution with the Schwarzian boundary condition at $R \to \infty$. Expanding the above general solution for $R \to \infty$ we find

$$Z_0 = \lim_{R\to\infty} Z(R,\sigma) = -i\frac{\sqrt{\pi}}{2}\int_{-\infty}^{\infty} d\nu\, e^{-\nu\sigma}a_\nu. \tag{122}$$

Notice that the $b_\nu$ coefficients do not play any role, since the Bessel function with positive argument goes as $e^{-4R}$. The function $Z_0$ is given by Schwarzian partition function,

$$Z_0 = \left(\frac{1}{2Ce^\sigma}\right)^{3/2} e^{\pi^2 e^{-\sigma}}, \tag{123}$$

Expanding this in $e^\sigma$ fixes the coefficients $a_\nu$ and after resumming using the multiplicative theorem for the Bessel $K_s(z)$ functions,

$$\alpha^{-s}K_s(\alpha z) = \sum_{n=0}^{\infty} \frac{(-1)^n}{2^n n!}(\alpha^2 - 1)^n z^n K_{s+n}(z), \tag{124}$$

we find the solution with the boundary condition (122) to be,

$$Z(R,\sigma) = i\frac{1}{\sqrt{2\pi C^3}}\frac{R^{3/2}e^{-2R-\sigma/2}}{Re^\sigma + \pi^2}K_2\left(-2\sqrt{R^2+\pi^2 Re^{-\sigma}}\right)$$
$$+ \int_0^\infty dv e^{-v\sigma}b_v\sqrt{R}e^{-2R}K_{v+1/2}(2R). \qquad (125)$$

The first term is precisely the deformed Schwarzian partition function found in [9]. The second term is there because the boundary condition at $R \to \infty$ is not enough to fully fix the solution. They are non-perturbative corrections to the partition function, discussed in 4. In that same section a proposal is presented how to fix, or at least partially, the $b_v$. In particular, by requiring $Z(R,\sigma)$ to be real. We know that $K_s(z)$ is real for $z > 0$ and since $R > 0$, we need $b_v$ to be complex in general. The Bessel $K_s(z)$ functions have a branch cut at the negative real axis and furthermore for integer $s$ we have,

$$K_s(-z) = (-1)^s K_s(z) + (\log(z) - \log(-z))I_s(z) \Rightarrow K_2(-z) = K_2(z) - i\pi I_2(z), \qquad (126)$$

where we used $z > 0$ and real after the implication arrow. Notice that here we also picked a particular branch of the logarithm so that $\log(-z) = \log(z) + i\pi$. This choice is motivated by the fact that as $R \to \infty$ the density of states of the corresponding partition function is positive. Consequently, to make $Z(R,\sigma)$ real we need the imaginary part of $b_v$, $b_v^{\text{Im}}$, to satisfy.

$$\frac{1}{\sqrt{2\pi C^3}}\frac{R^{3/2}e^{-2R-\sigma/2}}{Re^\sigma + \pi^2}K_2\left(2\sqrt{R^2+\pi^2 Re^{-\sigma}}\right) + \int_0^\infty dv e^{-v\sigma}b_v^{\text{Im}}\sqrt{R}e^{-2R}K_{v+1/2}(2R) = 0. \quad (127)$$

But this is the same matching as we did to implement the boundary condition (122), up to some signs. In fact, picking $b_v^{\text{Im}} = -(-1)^v a_v$ does the job and we get

$$Z(R,\sigma) = \sqrt{\frac{\pi}{2C^3}}\frac{R^{3/2}e^{-2R-\sigma/2}}{Re^\sigma + \pi^2}I_2\left(2\sqrt{R^2+\pi^2 Re^{-\sigma}}\right) + \tilde{Z}(R,\sigma), \qquad (128)$$

where

$$\tilde{Z}(R,\sigma) = \int_{-\infty}^\infty dv e^{-v\sigma}\sqrt{R}e^{-2R}c_v K_{1/2+v}(2R) \qquad (129)$$

with $c_v$ real. Going back to the $\lambda$ and $\beta$ variables, we find

$$Z_\lambda(\beta) = \sqrt{\frac{\pi}{2\lambda}}\frac{\beta e^{-\frac{\beta}{4\lambda}}}{\beta^2 + 16C\pi^2\lambda}I_2\left(\frac{1}{4\lambda}\sqrt{\beta^2 + 16C\pi^2\lambda}\right) + \tilde{Z}(\beta,\lambda). \qquad (130)$$

If one insists on getting a partiton function as a solution, i.e a solution that can be written as a sum over energies weighted by some Boltzmann factor, we can find solution in a simpler way. The ansatz is then

$$Z_\lambda(\beta) = \sum_E g(\lambda)e^{-\beta \mathcal{E}_\lambda(E)}. \qquad (131)$$

Plugging this in the differential equation (119) we precisely find the energy levels in (2) and $g(\lambda) = 1$, i.e. the density of states is not changed under the flow. If we consider a continuous spectrum we thus find (87).

# C  Details about regularization

**Some explicit perturbative calculations for $K[z(u)]$**

Since the discussion is section 3.5 is mostly formal, in this appendix we will compute the finite cutoff partition function to leading order in the cutoff $\varepsilon$. The unrenormalized quantities are $L = \beta/\varepsilon$ and $\phi_b = \phi_r/\varepsilon$. We want to reproduce the answer from WdW or $T\bar{T}$ which is given in (86). Expanding at small $\varepsilon$ gives

$$\log Z_{T\bar{T}} = \frac{2\pi^2\phi_r}{\beta} + \frac{3}{2}\log\left(\frac{\phi_r}{\beta}\right) - \varepsilon^2\left(\frac{2\phi_r\pi^4}{\beta^3} + \frac{5\pi^2}{\beta^2} + \frac{15}{8\phi_r\beta}\right) + \mathcal{O}(\varepsilon^4) \tag{132}$$

We want to reproduce the $\varepsilon^2$ term evaluating directly the path integral over the mode $z(u)$. Removing the leading $1/\varepsilon^2$ divergence we need to compute

$$Z_{\text{JT}}[\varepsilon] = \int \frac{\mathcal{D}z}{SL(2,\mathbb{R})} e^{\int_0^\beta du\,\phi_r K_2} e^{\varepsilon\int_0^\beta du\,\phi_r K_3 + \varepsilon^2\int_0^\beta du\,\phi_r K_4 + \cdots}, \tag{133}$$

where $K_2[z(u) = \text{Sch}(z,u)$ gives the leading answer and $K_3[z(u)]$ and $K_4[z(u)]$ are both given in (57) and contribute to subleading order. This integral is easy to do perturbatively. First we know that the expectation value of an exponential operator is equal to the generating function of connected correlators. Then any expectation value over the Schwarzian theory gives

$$\log\left\langle e^{\varepsilon\mathcal{O}[z]}\right\rangle_{\text{Sch}} = \log Z_0 + \varepsilon\langle\mathcal{O}[z]\rangle + \frac{\varepsilon^2}{2}\langle\mathcal{O}[z]\mathcal{O}[z]\rangle_{\text{conn}} + \cdots. \tag{134}$$

Using this formula we can evaluate the logarithm of the partition function to order $\varepsilon^2$ in terms of $K_3$ and $K_4$ as

$$\log Z_{\text{JT}} = \log Z_{\text{Sch}} + \varepsilon\int_0^\beta du\,\phi_r\langle K_3\rangle + \frac{\varepsilon^2}{2}\int_0^\beta dudu'\,\langle K_3 K_3'\rangle + \varepsilon^2\int_0^\beta du\,\phi_r\langle K_4\rangle + \mathcal{O}(\varepsilon^3). \tag{135}$$

The first correction is $K_3 = -i\partial_u\text{Sch}(z,u)$, which is a total derivative. This guarantees that, for a constant dilaton profile, the first two terms vanish since $\int du\langle K_3\rangle = 0$ and $\int\int dudu'\,\langle K_3 K_3'\rangle = 0$. The second correction is

$$K_4 = -\frac{1}{2}\text{Sch}(z,u)^2 + \partial_u^2\text{Sch}(z,u) \tag{136}$$

Then, since the second term in $K_4$ is a total derivative it can be neglected, giving

$$\log Z_{\text{JT}} = \log Z_{\text{Sch}} - \frac{\varepsilon^2}{2}\phi_r\int\langle\text{Sch}(z,u)^2\rangle + \mathcal{O}(\varepsilon^3). \tag{137}$$

Using point-splitting we can regulate the Schwarzian square. Schwarzian correlators can be obtained using the generating function. The one-point function is

$$\langle\text{Sch}(z,u)\rangle = \frac{1}{\beta}\partial_{\phi_r}\log Z = \frac{2\pi^2}{\beta^2} + \frac{3}{2\phi_r\beta} \tag{138}$$

The two point function is given by

$$\langle\text{Sch}(z,u)\text{Sch}(z,0)\rangle = -\frac{2}{\phi_r}\langle\text{Sch}(z,0)\rangle\delta(u) - \frac{1}{\phi_r}\delta''(u) + \langle:\text{Sch}(z,u)^2:\rangle \tag{139}$$

where we define the renormalized square Schwarzian expectation value as

$$\langle:\text{Sch}(z,u)^2:\rangle = \frac{4\pi^4}{\beta^4} + \frac{10\pi^2}{\beta^3\phi_r} + \frac{15}{4\beta^2\phi_r^2}. \tag{140}$$

This term only gives the right contribution matching the term in the $T\bar{T}$ partition function

$$\frac{\varepsilon^2}{2}\phi_r \int \langle : \mathrm{Sch}(z,u)^2 : \rangle = \varepsilon^2 \Big( \frac{2\phi_r \pi^4}{\beta^3} + \frac{5\pi^2}{\beta^2} + \frac{15}{8\phi_r \beta} \Big). \tag{141}$$

If evaluating $K_4[z(u)]$ without using the point-splitting procedure prescribed in section 3.5 then one naviely evaluates (140) at identical points. The divergent contributions can precisely be eliminated with the point-splitting prescription (77).

**Why derivatives of the Schwarzian don't contribute to the partition function**

Here we discuss in more detail why terms in $K[z(u)]$ containing derivatives of the Schwarzian do not contribute to the partition function (with constant dilaton value $\phi_r$) after following the point-splitting procedure (77). As mentioned in section 3.5 the schematic form of Schwarzian correlators is given by

$$\left( \frac{\delta}{\delta j(u_1)} \cdots \frac{\delta}{\delta j(u_n)} Z_{\mathrm{Sch}}[j(u)] \right)\bigg|_{j(u)=\phi_r} = a_1 + a_2 [\delta(u_{ij})] + a_3 [\partial_u \delta(u_{ij})] + \dots, \tag{142}$$

where the derivatives in the $\delta$-function terms above come by taking functional derivatives of the term $\exp\left( \int_0^\beta du \frac{j'(u)^2}{2j(u)} \right)$ in $Z_{\mathrm{Sch}}[j(u)]$. After following the point-splitting prescription (77) none of the functional derivatives of the form (142) that we will have to consider in the expansion of the exponential will be evaluated at identical points and therefore (142) will not contain terms containing $\delta(0)$ or its derivatives.

Consequently, note that when series-expanding the exponential functional derivative in (75), terms that contain derivatives in $\mathcal{K}\left[ \partial_u \frac{\delta}{\delta j(u)} \right]$ would give terms with contributions of the form

$$\int_0^\beta du_1 \cdots \int_0^\beta du_a \cdots \int_0^\beta du_N \left( \dots \partial_{u_a} \frac{\delta}{\delta j(u_a)} \dots Z_{JT}[j(u)] \right)\bigg|_{j(u)=\phi_r} =$$

$$= \int_0^\beta du_1 \cdots \int_0^\beta du_a \cdots \int_0^\beta du_N \left[ a_2 [\partial_u \delta(u_{ai})] + a_3 [\partial_u^2 \delta(u_{ai}), \partial_u \delta(u_{ai}) \partial_u \delta(u_{ak})] + \dots \right]$$

$$= 0, \tag{143}$$

where we note that $a_1$ vanishes after taking the derivative $\partial_{u_a}$.

In the second to last line we have that $a_2 [\partial_u \delta(u_{ai})]$ contains first order derivatives in $\delta(u_{ai})$ and $a_3 [\partial_u^2 \delta(u_{ai}), \partial_u \delta(u_{ai}) \partial_u \delta(u_{ak})]$ contains second-order derivatives acting on $\delta$-functions involving $u_a$. Since the functions above only contain $\delta$-functions involving other coordinates than $u_a$, all terms in the integral over $u_a$ vanish after integration by parts; consequently, the last line of (143) follows. Note that if we consider dilaton profiles that are varying $\phi_r(u)$ such derivative of $\delta$-function in fact would contribute after integration by parts. Consequently, it is only in the case of constant dilaton where such derivative terms do not give any contribution.

A very similar argument leads us to conclude that all other terms containing derivatives of $\delta$-functions in (142), vanish in the expansion of the exponential functional derivative from (75) when the $\delta$-function is evaluated at non-coincident points. Therefore, since the term $\exp\left( \int_0^\beta du \frac{j'(u)^2}{2j(u)} \right)$ only gives rise to terms containing derivatives of $\delta(u)$, this term also does not contribute when evaluating (75).

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
