# Peer review of "JT gravity at finite cutoff"

_SciPost Physics, doi:SciPost Phys. 9, 023 (2020)_

## Round 1 · Referee Report · Anonymous (Referee 1) · 2020-7-10

Report

This is a remarkable paper addressing issues about imposing a finite-cutoff in the JT gravity in a finite-cutoff setup. They compare the methods from the evaluation of the Wheeler-DeWitt functional and the gravitational path integral in the bulk, obtaining agreement in the boundary.

This paper illustrates new perspectives among JT gravity, TTbar deformation, and the information paradox setup discussed recently in the high energy theory community.

A particularly important problem related to this story, and I am particularly interested in is the possible finiteness of entanglement entropy in the holographic cutoff AdS in higher dimensions. The problem appears in higher dimensions, but I am curious if this also emerges in this 1+1 dimensional bulk, and I believe that this might also be related to the information problem in the setup of JT gravity. I look forward to the authors' comments on it.

---

## Round 1 · Referee Report · Anonymous (Referee 2) · 2020-7-20

Strengths

  1. Two concrete and detailed calculations of the bulk wavefunction/partition function are provided and matched with a previously existing boundary proposal, with many subtleties clearly elaborated.

  2. A novel proposal for fixing the complex energy levels of TTbar-deformed theories is provided.

Report

This paper is a wonderful addition to the TTbar literature and provides the first quantum-mechanical check of the "1d TTbar deformation" = "finite cutoff AdS2" proposal, by matching formulas derived from a Dirichlet cutoff bulk to previously derived formulas in the boundary theory. Several complementary issues are discussed, with a potential fix to the complexification of energy levels being a particularly important one.

Something that is not addressed is the connection to mixed boundary conditions at the usual AdS boundary, as detailed by Guica and Monten. There, the equivalence between TTbar deformations and Dirichlet conditions at finite cutoff in AdS was shown to be true classically (and expected to differ quantum mechanically), while the results of this paper suggest the two are equivalent even quantum mechanically. Is there an explanation of this seeming discrepancy?

---

## Round 1 · Referee Report · Anonymous (Referee 3) · 2020-8-2

Strengths

1-originality
2-importance of the results

Report

This paper concerns the study of the finite cut-off partition function of 2D Jackiw-Teitelboim gravity.
Two different methods are adopted; an exact evaluation of the functional Wheeler-DeWitt wave in radial quantization and the direct computation of the Euclidean path integral.
Both techniques lead to results which precisely match the partition function in the Schwarzian theory deformed by the 1D analogue of the TTbar deformation of 2D CFTs. The paper is very well written and addresses critical issues related to the interpretation of the TTbar deformation within the AdS/CFT framework.

The main result corresponds to the chain of equalities (1.12). Section 4 is particularly inspiring as two different types of non-perturbing corrections to the partition function are discussed.
The presence, in the partition function, of contributions coming from the "contracting branch" is fascinating and it will undoubtedly trigger further work.
In conclusion, this is an excellent piece of work that contains many original comments and opens the discussion on many challenging questions. I have no hesitation in recommending this document for publication.

Requested changes

1- After eq. B1: seperation ->separation

---

## Round 2 · Author Response

We thank the referees for their comments and questions. We will reply to each referee below.
Report 1:
We thank the referee for her/his kind words and question about holographic entanglement entropy. We are not entirely sure what entanglement entropy is meant here by the referee. In higher dimensions ($d>1$) there is space and one can compute the holographic entanglement entropy of some spatial region. In the bulk (say, for instance AdS$_3$), the EE in TTbar deformed theories then corresponds to the area of an RT surface extending to some finite radial coordinate, which makes it finite. In the present case there is no space or RT surfaces in the bulk and therefore also no holographic entanglement entropy.
Report 2:
We thank the referee for her/his kind words and question. Regarding the Guica/Monten story; in this manuscript we have not attempted to compute the partition function with that definition of the $T\bar{T}$ deformed theory and so we cannot make the comparison in the present work. It would certainly be very interesting to work that out in detail and compare the two prescriptions on a quantum mechanical level.
Report 3:
We thank the referee for her/his kind words and spotting the typo. We have fixed it.
We hope that with this we have answered the questions raised by the referees in a satisfactory manner.
Sincerely yours,
Luca V. Iliesiu, Jorrit Kruthoff, Gustavo J. Turiaci and Herman Verlinde.

---

## Editorial Decision

published